# A Solver-Free Training Method for Predict-then-Optimize

**Beichen Wan** [1]   **Mo Liu** [1]

## Abstract

We propose a scalable method for training prediction (machine learning) models in the predict-then-optimize paradigm, where model outputs serve as coefficients for a subsequent linear optimization task. Directly minimizing the empirical decision regret is intractable for linear programming and combinatorial optimization since the decision mapping is piecewise constant, and the gradients are zero almost everywhere. While existing methods address this by smoothing the differentiation process, they suffer from scalability issues, since a computationally expensive solver call is required for every gradient evaluation. To address this, we propose a decision-focused learning pipeline based on a measure transformation principle, which yields a new surrogate loss that is completely optimization-solver-free during training. We establish theoretical guarantees, including Fisher consistency and excess risk bounds. Empirically, our method achieves decision quality competitive with state-of-the-art methods while reducing training time by orders of magnitude.

## 1. Introduction

We consider a contextual stochastic linear optimization setting where a decision-maker observes contextual features $x \in \mathbb{R}^p$ and must choose a decision vector $w$ from a known, non-empty, compact feasible region $\mathcal{S} \subseteq \mathbb{R}^d$. The feasible region $\mathcal{S}$ is allowed to be general, encompassing polyhedral, discrete, or strongly convex sets. The realized cost of a decision is linear in an unknown parameter vector $y \in \mathbb{R}^d$, which is not observed at the time of decision. While the joint distribution of $(x, y)$ is unknown, we assume that $y$ is statistically dependent on $x$; we denote this unknown joint distribution by $P$. For an observed feature input $x \in \mathbb{R}^p$, the decision-maker solves:

$$\min_{w \in \mathcal{S}} \mathbb{E}_P[Y^\top w \mid x] = \min_{w \in \mathcal{S}} \mathbb{E}_P[Y \mid x]^\top w. \qquad (\mathcal{P}1)$$

Since $\mathcal{S}$ is a general feasible region, problem $(\mathcal{P}1)$ represents a broad class of decision-making problems, depending on the specific structure of $\mathcal{S}$, including *vehicle routing, inventory management, portfolio optimization, bipartite matching*, and *label ranking*.

Solving $(\mathcal{P}1)$ requires knowledge of the conditional expectation of the cost vector $\mathbb{E}_P[Y|x]$ for any observed $x \in \mathbb{R}^p$. To predict $\mathbb{E}_P[Y|x]$, we consider a data-driven setting where the decision-maker has access to historical observations $(x_i, y_i)_{i=1}^n$. Let $w^*(\hat{y}) = \arg\min_{w \in \mathcal{S}} \hat{y}^\top w$ denote the optimization solution of the downstream decision-making problem for an input cost vector $\hat{y}$. Note that $w^*(\hat{y})$ is a known function for any input prediction $\hat{y}$ since we have full knowledge of the feasible region $\mathcal{S}$. If there is a prediction $f$ of $\mathbb{E}_P[Y \mid x]$, then for any $x \in \mathbb{R}^p$, the decision-maker can use the plug-in solution $w^*(f(x))$ to solve the Problem $(\mathcal{P}1)$, i.e.

$$w^*(f(x)) \in \arg\min_{w \in \mathcal{S}} f(x)^\top w.$$

Assuming a consistent tie-breaking rule ensures uniqueness (e.g., by ordering solutions), the learning objective of predict-then-optimize becomes finding a prediction function $f$ from a hypothesis class $\mathcal{F} : \mathbb{R}^p \to \mathbb{R}^d$ that minimizes the excess decision regret:

$$\min_{f \in \mathcal{F}} \mathbb{E}_{(X,Y) \sim P} \left[ Y^\top w^*(f(X)) - Y^\top w^*(Y) \right] \qquad (\mathcal{P}2)$$

Since $Y^\top w^*(Y)$ is independent of $f$, minimizing the expected regret is equivalent to minimizing the expected decision cost $\mathbb{E}_{(X,Y) \sim P}[Y^\top w^*(f(X))]$. We define the decision loss with respect to a realized cost vector $y$ and a predictor $\hat{y}$ to be $\ell_{\text{decision}}(\hat{y}, y) := y^\top w^*(\hat{y}) - y^\top w^*(y)$.

The methods for solving Problem $(\mathcal{P}2)$ can be classified into two classes. The first class of methods is called decision-blind methods, which use pure prediction loss (e.g., the squared loss $\ell_2$) to find $f$ that estimates $\mathbb{E}_P[Y \mid x]$. The second class is called decision-focused learning (DFL) methods, which not only utilizes prediction quality but also leverages decision quality to train $f$. Decision-blind methods

[1]Department of Statistics and Operations Research, University of North Carolina at Chapel Hill, NC, USA. Correspondence to: Beichen Wan <bcwan@ad.unc.edu>, Mo Liu <mo_liu@unc.edu>.

*Proceedings of the 43rd International Conference on Machine Learning*, Seoul, South Korea. PMLR 306, 2026. Copyright 2026 by the author(s).

provide a simple yet efficient training pipeline, whereas DFL methods enable the training process to better align with the true decision-making goal.

A fundamental challenge arises in designing DFL methods for Problem ($\mathcal{P}2$) when the feasible region $\mathcal{S}$ is polyhedral or discrete. In this case, the loss function $\ell_{\text{decision}}(\cdot, y)$ will be piecewise constant, hence non-convex and non-continuous, and its gradient will be either zero or undefined on the decision borders. As a consequence, first-order methods cannot be applied since the gradients are uninformative. To address this, past DFL methods have focused on smoothing the differentiation process to make first-order methods effective again. Despite the success of these DFL methods in particular tasks, they all require calling the optimization solver each time when evaluating gradients. When the underlying optimization problem is complex and time-consuming to solve (e.g., large-scale mixed-integer linear programming), these methods become computationally prohibitive.

In this paper, we address this computational issue by developing a solver-free surrogate loss, which is developed by the measure transformation of the source distribution. This transformation enables decision-blind training objectives to better align with the downstream decision-making goal. Our approach retains the computational efficiency of lightweight decision-blind surrogates while leveraging the geometric structure of linear optimization. On the theory side, our framework connects to least squares regression naturally, and thus inherits its favorable properties, including Fisher consistency and excess regret bounds.

## 1.1. Our Contributions

- We design a training method for prediction models in the predict-then-optimize paradigm that is both solver-free and aligned with the downstream decision cost.

- We apply measure transformation to derive a new solver-free surrogate loss, called the Weight Integrated Spherical Error (WISE). To our knowledge, we are the *first* to study decision-focused learning under measure transformation.

- We provide theoretical guarantees for the WISE surrogate loss, including Fisher consistency, calibration bounds, and finite sample excess regret bounds.

- We show that our method yields competitive performance with respect to other state-of-the-art methods in both well-specified and misspecified cases.

## 1.2. Related Literature

The predict-then-optimize framework has been extensively studied in the data-driven decision-making literature (El-machtoub & Grigas, 2022; Bertsimas & Kallus, 2020; Mandi et al., 2024; Sadana et al., 2025). To align predictive models with the decision-making objectives, researchers have developed decision-focused learning methods, also referred to as differentiable optimization or end-to-end learning. We organize representative and related papers from three angles: differentiation strategies, scalability and approximation, and theoretical analysis, as follows:

**Differentiation Strategies** In this area, existing approaches can be categorized by how they differentiate through the downstream optimization problem. When the downstream problem has analytical optimality conditions (e.g., KKT for quadratic programming), implicit differentiation methods allow for backpropagating gradients through the optimizer (Donti et al., 2017; Amos & Kolter, 2017; Agrawal et al., 2019). On the other hand, for problems with polyhedral or discrete feasible regions, researchers have developed convex surrogate losses (Elmachtoub & Grigas, 2022), randomized smoothing (Berthet et al., 2020), regularized objective (Wilder et al., 2019; Ferber et al., 2020; Mandi & Guns, 2020), and continuous interpolations of the loss function (Pogančić et al., 2019; Huang & Gupta, 2024).

**Scalability and Approximation** While effective, the aforementioned methods can be computationally prohibitive for large-scale optimization problems, since every gradient evaluation requires calling the optimization solver. To alleviate this, recent works have proposed warmstarting the solving and learning via relaxation (Mandi et al., 2020), projecting to a low-dimension feasible region (Wang et al., 2020), fitting a local convex surrogate (Shah et al., 2022), fitting a global landscape surrogate (Zharmagambetov et al., 2023), and approximating the solver via unrolled neural networks (Cristian et al., 2025). (Tang & Khalil, 2024a) proposed a scalable method for binary linear programming that utilizes the projection of the predicted cost vector to the optimal normal cone. (Berden et al., 2026) proposed a solver-free training method for linear optimization by comparing a known optimal solution with its adjacent vertices on the feasible polytope, which is related to contextual inverse optimization settings where past optimal decisions are observed (Mishra et al., 2024). However, these solver-free or approximately solver-free methods still rely on observed decisions, precomputed optimal solutions, or local geometric information around them. Therefore, when only true cost vectors are observed and past optimal decisions are unavailable, they do not provide an entirely solver-free training pipeline.

**Theoretical Analysis** On the theoretical front of contextual linear optimization, (El Balghiti et al., 2022; Liu & Grigas, 2021; Ho-Nguyen & Kılınç-Karzan, 2022) derived generalization bounds and risk bounds by leveraging the

geometric properties. (Hu et al., 2022) demonstrated that a simple decision-blind $\ell_2$ loss can achieve a faster rate than DFL methods under well-specification, suggesting that DFL methods are mainly useful under misspecification. (Lan et al., 2026; Elmachtoub et al., 2025; 2023) further study the impact of model misspecification of general stochastic optimization by analyzing the asymptotic convergence of decision-blind and DFL estimators.

**Applications** The ideas and methodologies of DFL have been applied across a wide range of areas, including revenue management (Ban & Rudin, 2019; Ban & Keskin, 2021; Liu et al., 2023a) and capacity allocation (Er & Liu, 2025). DFL has also motivated new analyses and tools for classical statistical learning questions, such as data collection (Liu et al., 2023b; Wan et al., 2026), data efficiency (Gupta & Rusmevichientong, 2021; Gupta et al., 2024), and optimal transport (Liu & Liu, 2026).

### 1.3. Notation

In the paper, we use the following notions. We use $\|\cdot\|$ to denote the $\ell_2$-norm. For a given prediction model $f \in \mathcal{F}$, we define its risk for a given loss $\ell$ under probability measure $P$ as

$$R_P^\ell(f) = \mathbb{E}_{(X,Y)\sim P}[\ell(f(X), Y)].$$

We further plug in the conditional expectation and obtain $R_P^* := R_P^{\ell_{\text{decision}}}(\mathbb{E}_P[Y|x])$, which denotes the minimum decision risk.

## 2. Decision Focused Learning via Measure Transformation

In this section, we derive our proposed surrogate loss function for the training. Unlike previous approaches that design surrogates by smoothing the optimization oracle or bounding the decision regret, we aim for a method that is completely solver-free and exploits the geometry of linear optimization.

Standard decision-blind methods like minimizing $\ell_2$ loss are often suboptimal for "predict-then-optimize" because there exists a misalignment between the learning objective (prediction accuracy) and the downstream task objective (decision regret). This misalignment arises because decision-blind losses treat the target space as Euclidean and ignore the geometry induced by the downstream optimization problem. In contextual linear optimization, the optimal decision $w^*(\hat{y})$ is determined primarily by the *direction* of the cost vector $\hat{y}$, and the *magnitude* of the regret is scaled by the norm of the true cost $\|y\|$.

Recall that the decision loss is defined as $\ell_{\text{decision}}(\hat{y}, y) = y^\top w^*(\hat{y}) - y^\top w^*(y)$. This loss function exhibits two geometric properties:

- Scale invariant: For any scalar $\alpha > 0$, cost prediction $\hat{y}$, and true cost vector $y$, the decision loss is invariant with respect to the scaling of the prediction $\hat{y}$, i.e.,

$$\ell_{\text{decision}}(\alpha\hat{y}, y) = \ell_{\text{decision}}(\hat{y}, y).$$

- Loss scaling: For any scalar $\beta > 0$, cost prediction $\hat{y}$, and true cost vector $y$, the decision loss scales with the true cost $y$, i.e.,

$$\ell_{\text{decision}}(\hat{y}, \beta y) = \beta\ell_{\text{decision}}(\hat{y}, y).$$

The scale invariant property comes from the identity $w^*(\hat{y}) = w^*(\alpha\hat{y})$. Geometrically, solving a linear optimization problem is equivalent to finding a decision vector $w$ that is negatively aligned with $\hat{y}$, and the magnitude of $\hat{y}$ does not affect the solution. See Appendix A for a detailed proof and discussion. The high-level idea of the proof is to use the fact that

$$\begin{aligned}
\ell_{\text{decision}}(\alpha\hat{y}, y) &= y^\top w^*(\alpha\hat{y}) - y^\top w^*(y) \\
&= y^\top w^*(\hat{y}) - y^\top w^*(y) \\
&= \ell_{\text{decision}}(\hat{y}, y),
\end{aligned}$$

This implies that learning a prediction function $f$ should focus on recovering the direction of $\mathbb{E}_P[Y \mid x]$, rather than its magnitude.

On the other hand, the loss scaling property follows directly from the definition of the linear optimization objective, i.e.,

$$\begin{aligned}
\ell_{\text{decision}}(\hat{y}, \beta y) &= (\beta y)^\top w^*(\hat{y}) - (\beta y)^\top w^*(\beta y) \\
&= \beta \left( y^\top w^*(\hat{y}) - y^\top w^*(\beta y) \right) \\
&= \beta \left( y^\top w^*(\hat{y}) - y^\top w^*(y) \right) \\
&= \beta\ell_{\text{decision}}(\hat{y}, y),
\end{aligned}$$

where the third equation we used the identity $w^*(\beta y) = w^*(y)$ shown in the previous discussion. This indicates that the decision loss is not uniform across the distribution; suboptimal decisions induce strictly higher regret when the true cost vector has a large scale. Thus, the learning objective should prioritize samples with a larger scale.

The above two properties of decision loss hold for any specific problem in the linear optimization class, regardless of the specific structure of the downstream optimization problem. However, existing standard decision-blind methods all fail to capture these two properties.

To rigorously address this, we propose a transformation of the underlying probability measure to align the decision-blind loss with the geometry of the optimization task. This method acts as a simple pre-processing step and is compatible with any decision-blind learner. Figure 1 illustrates

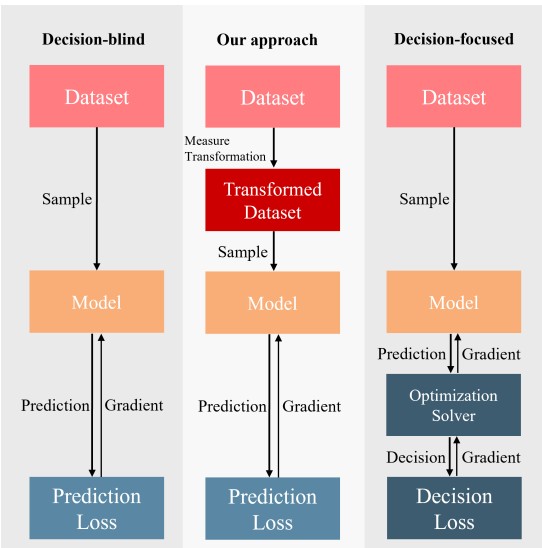

*Figure 1.* Schematic comparison of *training* pipelines. (Left) Decision-blind learning minimizes prediction error without considering the downstream task. (Right) Decision-focused methods incorporate the optimization solver into the training loop, requiring computationally expensive differentiation through the solver. (Mid) Our approach integrates the optimization structure by a measure transformation, enabling a completely solver-free training process that maintains the computational efficiency of decision-blind methods while achieving decision-aware performance.

our pipeline alongside standard decision-blind and decision-focused learning. Our method is completely solver-free yet effectively incorporates decision-focused components.

### 2.1. Measure Transformation

To enforce the geometric properties derived in the previous section, we propose a two-step transformation of the original probability measure $P$: a *reweighting* based on cost magnitude, and a *projection* onto the unit sphere.

We assume the cost vector is integrable ($\mathbb{E}_P[\|Y\|] < \infty$) and $\|Y\| > 0$ almost surely. Note that $\|y\| = 0$ indicates a degenerate case where the true cost vector is 0. In this case, any prediction induces a 0 decision loss, and hence provides no gradient information for linear optimization. Hence, we restrict our attention to probability measures $P$ supported on $\mathbb{R}^p \times \mathbb{R}^d \setminus \{0\}$ to avoid degeneracy and ensure the projection is well defined.

**Step 1: Reweighting Measure $P$.** To address the *Loss Scaling* property, we define a new probability measure $\tilde{Q}$ that reweights $P$ proportionally to the magnitude of the cost vector, i.e.,

$$d\tilde{Q}(x, y) \propto \|y\| dP(x, y).$$

More formally speaking, we define a measure $\tilde{Q}$ that is absolutely continuous with respect to $P$ ($\tilde{Q} \ll P$), and the

Radon-Nikodym derivative is defined as:

$$\frac{d\tilde{Q}}{dP}(x, y) = \frac{\|y\|}{C}. \qquad \text{(Step 1)}$$

where $C = \mathbb{E}_P[\|Y\|]$ is a normalizing constant since $\int \|y\| dP(x, y) = \mathbb{E}_P[\|Y\|]$. By assigning higher probability to samples with large cost magnitudes, $\tilde{Q}$ naturally aligns the learning objective with the true decision regret.

**Step 2: Projecting Measure $\tilde{Q}$ onto the Unit Sphere.** To address the *Scale Invariance* property (linear optimization solutions only depend on the direction of the cost vector), we project the reweighted measure onto the unit sphere $\mathbb{S}^{d-1}$. Let $T : \mathbb{R}^p \times \mathbb{R}^d \setminus \{0\} \to \mathbb{R}^p \times \mathbb{S}^{d-1}$ be the projection mapping and define $Q$ as the *push-forward measure* of $\tilde{Q}$ under $T$:

$$T(x, y) = (x, \frac{y}{\|y\|}), \quad Q = \tilde{Q} \circ T^{-1}. \qquad \text{(Step 2)}$$

In this way, we define $Q$ as the projected measure of $\tilde{Q}$ onto the unit sphere, which emphasizes the focus on directional learning.

In terms of random variables, this process is equivalent to sampling $(X, Y)$ from the reweighted measure $\tilde{Q}$ and normalizing the target to obtain $(X, Z) = (X, Y/\|Y\|)$. The resulting measure $Q$ not only assigns a correct weight for each sample but also isolates the directional component of the cost, forcing the learner to focus on the geometry of the optimization landscape. Figure 2 illustrates the effect of the measure transformation on a 2-dimensional Gaussian distribution.

To see that our proposed measure transformation indeed captures the geometric property of the decision loss function, we establish that the decision risks under measures $P$ and $Q$ are equivalent up to a constant multiplicative factor, formalized in the following proposition.

**Proposition 2.1** (Decision Risk Equivalence under Measure Transformation)**.** *Let $P$ be a probability measure on $\mathbb{R}^p \times \mathbb{R}^d$ satisfying $\|Y\| > 0$ almost surely and $C < \infty$, where $C$ is defined in Step 1. Let $Q$ be the measure defined by the reweighting and projection procedure in Step 1 and Step 2.*

*Then, for any measurable function $f : \mathbb{R}^p \to \mathbb{R}^d$, the following identity holds:*

$$\mathbb{E}_{(X,Z) \sim Q}[\ell_{\text{decision}}(f(X), Z)] = \frac{1}{C} R_P^{\ell_{\text{decision}}}(f).$$

This proposition further implies that the decision-risk minimizers under $P$ and $Q$ coincide, since the two decision risks differ only by a positive constant factor. Therefore, if a surrogate loss is Fisher consistent (i.e., minimizing the surrogate risk also minimizes the decision risk) on the original

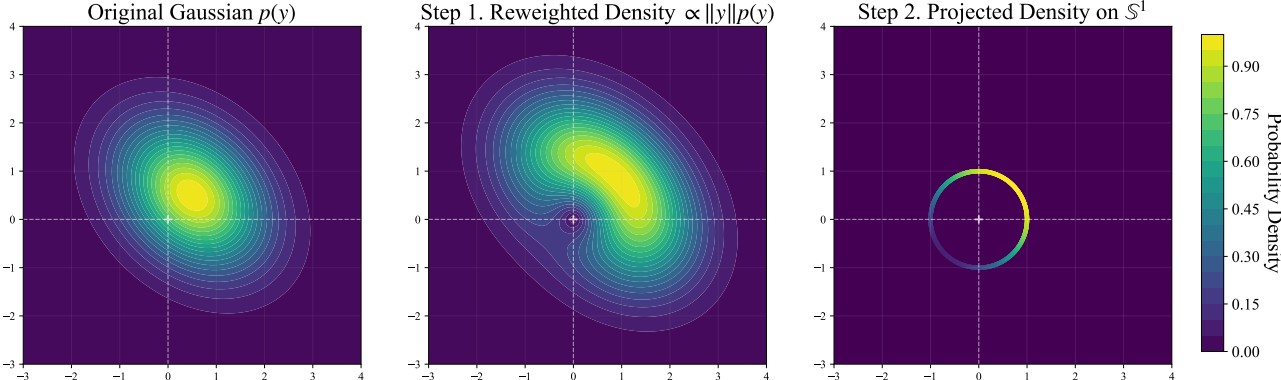

*Figure 2.* Illustration of the proposed probability measure transformation process, here we ignore the dependencies on $X$ and only focus on $Y$ since the transformation is mainly about the marginal distribution of $Y$. (Left) The original source distribution $Y \sim \mathcal{N}(\mu, \Sigma)$ with $\mu = [0.5, 0.5]^\top$ and $\Sigma = [1, -0.3; -0.3, 1]$. (Middle) The intermediate reweighted density $\tilde{Q}$, proportional to $\|y\|p(y)$, which shifts probability mass based on the norm. (Right) The final projected density $Q$ on the unit sphere $\mathbb{S}^1$.

measure, it remains Fisher consistent on the transformed probability measure.

Note that both Step 1 and Step 2 are necessary for the Fisher consistency, a further discussion on why Step 1 and Step 2 should *not* be applied in isolation is provided in Appendix B.

### 2.2. $\ell_{\text{WISE}}$: $\ell_2$ Loss under Measure Transformation

Having defined the transformation $P \to Q$, we advocate minimizing the risk on $Q$ to capture the decision loss landscape, rather than the standard risk on $P$. However, explicitly transforming the dataset to match $Q$ can be computationally cumbersome. To further simplify the working pipeline, we derive the equivalent surrogate loss that operates directly on $P$. This derivation provides a practical advantage by yielding a ready-to-use loss function that can be applied to the original dataset without transformation.

Inspired by Proposition 2.1, a natural strategy to derive a Fisher consistent surrogate loss is to adapt a Fisher consistent surrogate from $Q$ to the original measure $P$. We choose to work with $\ell_2$ since it is the most canonical surrogate with desirable analytical properties. Most importantly, the $\ell_2$ loss is Fisher consistent as it yields $\mathbb{E}[Y|x]$ as the Bayes risk minimizer. (In contrast, the mean absolute error (MAE) is not Fisher consistent because its Bayes risk minimizer is the conditional median).

By formulating the $\ell_2$ risk on the transformed measure $Q$, we derive a new surrogate loss called the Weight Integrated Spherical Error ($\ell_{\text{WISE}}$), as detailed in Proposition 2.2.

**Proposition 2.2** (The Equivalent Form of $R_Q^{\ell_2}$ under $P$ Measure)**.** *Under the same assumption in Proposition 2.1, for any measurable function $f : \mathbb{R}^p \to \mathbb{R}^d$, the following*

*identity holds:*

$$\mathbb{E}_{(X,Z)\sim Q} \left[ \|f(X) - Z\|^2 \right]$$
$$= \frac{1}{C} \mathbb{E}_{(X,Y)\sim P} \left[ \|Y\| \left\| f(X) - \frac{Y}{\|Y\|} \right\|^2 \right].$$

Recall that $C = \mathbb{E}_P[\|Y\|]$ is a normalizing constant independent of the estimator. Proposition 2.2 implies that minimizing the surrogate on the right-hand side of the above equation yields the same solution as $\ell_2$ risk minimization. We formally define this new surrogate loss as:

$$\ell_{\text{WISE}}(\hat{y}, y) = \begin{cases} \|y\| \cdot \left\| \hat{y} - \frac{y}{\|y\|} \right\|^2 & \text{if } \|y\| \neq 0, \\ 0 & \text{if } \|y\| = 0. \end{cases}$$

*Remark* 2.3. The $\ell_{\text{WISE}}$ function is well defined in the sense that it is continuous with respect to all $\hat{y}, y$ since $\lim_{\|y'\|\to 0} \ell_{\text{WISE}}(\hat{y}, y') = 0$. Consequently, the numerical stability issue when $\|y\|$ is close to zero does not influence the implementation since either thresholding the loss to zero or clamping $\|y\|$ to a pre-specified tolerance $\epsilon$ will fix the issue. Intuitively, the issue does not exist since a near-zero cost vector does not provide effective information on decision making. $\qquad\square$

Based on Remark 2.3, without loss of generality, we assume $\|y\|$ is strictly positive almost surely and write the corresponding loss function as $\ell_{\text{WISE}}(\hat{y}, y) = \|y\| \cdot \left\| \hat{y} - \frac{y}{\|y\|} \right\|^2$ in the remainder of the paper.

## 3. Theoretical Guarantees

In this section, we analyze the theoretical properties of the proposed $\ell_{\text{WISE}}$ loss. We impose the following standard

boundedness conditions to ensure the validity of the high-probability bounds:

**Assumption 3.1** (Boundedness). We assume the following uniform bound exists:

1. Bounded predictor: There exists a constant $B > 0$ such that $0 < \|f(x)\| \leq B$ for all $f \in \mathcal{F}$ almost surely.

2. Bounded feasible region: There exists a constant $B_{\mathcal{S}} > 0$ such that $\|w\| \leq B_{\mathcal{S}}$ for all $w \in \mathcal{S}$.

3. Bounded cost: There exists a constant $B_y > 0$ such that $0 < \|y\| \leq B_y$ almost surely.

These are standard assumptions (Liu & Grigas, 2021; Hu et al., 2022) which can be ensured, for example, by restricting the hypothesis class $\mathcal{F}$ or projecting outputs.

### 3.1. Analytical Properties and Fisher Consistency

Naturally, $\ell_{\text{WISE}}$ inherits nice analytical properties from $\ell_2$, formalized in the following proposition.

**Proposition 3.2.** *Under Assumption 3.1, for any fixed $y$, the function $\ell_{\text{WISE}}(\cdot, y)$ is continuously differentiable, convex, and satisfies the following properties:*

1. *Smoothness: $\ell_{\text{WISE}}$ is $2B_y$-smooth.*

2. *Local Lipschitz Continuity: $\ell_{\text{WISE}}$ is $L$-Lipschitz on the bounded domain, with constant $L = 2B_y(B + 1)$.*

3. *Boundedness: $\ell_{\text{WISE}}$ is bounded by $B_y(B + 1)^2$.*

By the properties in Proposition 3.2, the $\ell_{\text{WISE}}$ loss can be efficiently minimized using first-order methods.

In least squares minimization without any restrictions on the prediction class $\mathcal{F}$, a well-known result is that the Bayes risk minimizer is exactly the conditional expectation. When minimizing the $\ell_{\text{WISE}}$, a similar result holds, as shown in the following proposition.

**Proposition 3.3** (Bayes Risk Minimizer of $\ell_{\text{WISE}}$). *Suppose Assumption 3.1 holds. The Bayes risk minimizer of $\ell_{\text{WISE}}$ under $P$ measure is given by*

$$f^*(x) = \frac{\mathbb{E}_P[Y \mid x]}{\mathbb{E}_P[\|Y\| \mid x]}.$$

Proposition 3.3 shows that minimizing the $\ell_{\text{WISE}}$ risk yields a weighted conditional expectation. Note that $f^*(x)$ always lies within the unit ball, since $\|f^*(x)\| = \frac{\|\mathbb{E}_P[Y|x]\|}{\mathbb{E}_P[\|Y\| \mid x]} \leq 1$, where the inequality follows from Jensen's inequality.

Finally, we conclude with a Fisher consistency result.

**Theorem 3.4** (Fisher Consistency). *Suppose Assumption 3.1 holds. Let $f^*$ be the $\ell_{\text{WISE}}$ Bayes risk minimizer under measure $P$, as stated in Proposition 3.3, then $f^*$ also minimizes the decision risk, i.e.,*

$$f^* \in \arg \min_{f:\mathbb{R}^p \to \mathbb{R}^d} \mathbb{E}_{(X,Y) \sim P}[\ell_{\text{decision}}(f(X), Y)].$$

Theorem 3.4 establishes Fisher consistency without imposing any restrictions on the hypothesis class. In the next section, we study non-asymptotic risk bounds under finite samples and a structured prediction class.

### 3.2. Excess Regret Bound

In this section, we provide a finite-sample excess regret bound for $\ell_{\text{WISE}}$. We start the analysis with a calibration argument, given below:

**Proposition 3.5** (Excess Regret Calibration). *Let $f^*(x) = \frac{\mathbb{E}_P[Y|x]}{\mathbb{E}_P[\|Y\| | x]}$ be the true normalized conditional expectation (the Bayes risk minimizer of $\ell_{\text{WISE}}$). Suppose Assumption 3.1 holds. Then, for any predictor $f$, the excess decision regret satisfies the following bound:*

$$R_P^{\ell_{\text{decision}}}(f) - R_P^* \leq 2B_{\mathcal{S}} B_y^{\frac{1}{2}} \sqrt{R_P^{\ell_{\text{WISE}}}(f) - R_P^{\ell_{\text{WISE}}}(f^*)}.$$

This excess regret calibration method provides the same order of calibration using the $\ell_2$ surrogate (Hu et al., 2022) and the SPO+ surrogate (Liu & Grigas, 2021).

Next, we provide an excess risk bound based on a hypothesis class of bounded VC-type dimension. We first give the following definition and assumption used in (Hu et al., 2022), which is a generalization of VC-dimension to vector-valued functions.

**Definition 3.6** (VC-Linear-Subgraph Dimension). The VC-linear-subgraph dimension of a class of functions $\mathcal{F} \subseteq [\mathbb{R}^p \to \mathbb{R}^d]$ is the VC dimension of the sets $\mathcal{F}^\circ = \{\{(x, \beta, t) : \beta^\top f(x) \leq t\} : f \in \mathcal{F}\}$ in $\mathbb{R}^{p+d+1}$, i.e., the largest integer $\nu$ for which there exist $x_1, \ldots, x_\nu \in \mathbb{R}^p$, $\beta_1, \ldots, \beta_\nu \in \mathbb{R}^d$, $t_1 \in \mathbb{R}$, $\ldots$, $t_\nu \in \mathbb{R}$ such that

$$\{(\mathbb{1}[\beta_1^\top f(x_1) \leq t_1], \ldots, \mathbb{1}[\beta_\nu^\top f(x_\nu) \leq t_\nu]) : f \in \mathcal{F}\} = \{0, 1\}^\nu.$$

Equipped with the new definition and preliminary calibration result, we obtained the following excess regret bound for a bounded VC-dimension hypothesis class.

**Theorem 3.7** (Excess Regret Bound). *Suppose Assumption 3.1 holds and assume $f^* \in \mathcal{F}$. Let $\hat{f}_n \in \arg \min_{f \in \mathcal{F}} \frac{1}{n} \sum_{i=1}^n \ell_{\text{WISE}}(f(X_i), Y_i)$ denote the empirical risk minimizer, assume $\mathcal{F}$ is star-shaped at $f^*$, i.e.*

$(1 - \lambda)f + \lambda f^* \in \mathcal{F}$ *for all* $f \in \mathcal{F}$ *and* $\lambda \in [0, 1]$ *and the VC-linear-subgraph dimension of* $\mathcal{F}$ *is at most* $\nu$. *Then there exist constants* $C_0, C_1, C_2 > 0$ *depending only on the boundedness constants* $(B, B_y, B_{\mathcal{S}})$ *such that for all* $n \geq C_0$ *and all* $\delta \in (0, 1/2]$, *with probability at least* $1 - \delta$,

$$R_P^{\ell_{\mathrm{decision}}}(\hat{f}_n) - R_P^* \leq C_2 \sqrt{\frac{\nu \log(C_1 n) + \log(1/\delta)}{n}}.$$

This bound matches the classic $O(n^{-\frac{1}{2}})$ order in past literature (Hu et al., 2022; El Balghiti et al., 2022).

### 3.3. Faster Rates under Strongly Convex Level Sets

The convergence results in the previous section work for general feasible regions (e.g., a polyhedron). However, the rate can be improved if the feasible region has additional structures. In particular, we study the case where the feasible region can be defined as the level set of strongly convex functions, as stated in the assumption below:

**Assumption 3.8** (Strongly Convex Level Set). *Let* $g : \mathbb{R}^d \to \mathbb{R}$ *be a function that is* $\mu$-*strongly convex and* $L$-*smooth function where* $L \geq \mu > 0$. *Assume the feasible region is defined by* $\mathcal{S} = \{w \in \mathbb{R}^d : g(w) \leq r\}$ *for some* $r > g_{\min} = \min_w g(w)$.

This construction of a feasible region guarantees a smooth transition between prediction and decision. Typical examples for strongly convex level sets include bounded $\ell_2$ norm balls and ellipsoids; see (Liu & Grigas, 2021) for details.

**Corollary 3.9.** *Suppose that Assumption 3.1 and Assumption 3.8 hold. Further assume there exists a uniform constant* $\gamma > 0$ *such that* $\|\mathbb{E}_P[Y|x]\| \geq \gamma$ *for all* $x$ *almost surely. Then the excess decision regret satisfies the following calibration bound*

$$R_P^{\ell_{\mathrm{decision}}}(f) - R_P^* \leq M(R_P^{\ell_{\mathrm{WISE}}}(f) - R_P^{\ell_{\mathrm{WISE}}}(f^*)).$$

*where* $M = \frac{2B_y^2 L^2 \sqrt{2(r - g_{\min})}}{\gamma^2 \mu^{2.5}}$.

*Moreover, if the assumptions in Theorem 3.7 hold, then there exist positive constants* $C_0, C_1, C_2$ *depending only on the boundedness and geometry constants such that for any* $\delta \leq (nd + 1)^{-C_0}$, *with probability at least* $1 - C_1 \delta^\nu$,

$$R_P^{\ell_{\mathrm{decision}}}(\hat{f}_n) - R_P^* \leq C_2 \frac{\nu \log(1/\delta)}{n}$$

This $O(n^{-1})$ bound improves the convergence rate of the previous excess regret bound under a general feasible set. Moreover, our rate is faster than the $O(n^{-\frac{1}{2}})$ SPO+ surrogate risk bound under strongly convex level sets, derived by (Liu & Grigas, 2021).

## 4. Experiments

To empirically validate the performance of our method, we compare it against standard decision-blind surrogate and canonical DFL methods across a shortest path problem, a 2-dimensional knapsack problem, and a portfolio optimization problem. These tasks can represent the most common linear optimization settings, featuring either feasible regions with polyhedral sets, discrete sets, or general convex sets.

We consider learning from a linear prediction class, benchmarking the following methods: the two-stage decision-blind mean squared error (MSE), the SPO+ (Elmachtoub & Grigas, 2022), PFY (Berthet et al., 2020), and DBB(Pogančić et al., 2019). Note that DBB is equivalent to the forward mode of Perturbed Gradient (Huang & Gupta, 2024), the most recent proposed DFL surrogate. We also compare with scalable approaches, including CAVE (Tang & Khalil, 2024a) and LAVA (Berden et al., 2026), which are designed for polyhedral feasible regions.

Regarding evaluation metrics of a predictor, we used the normalized regret on the test set $\{(x_i, y_i)\}_{i=1}^{n_{\mathrm{test}}}$, defined as

$$\frac{\sum_{i=1}^{n_{\mathrm{test}}} \ell_{\mathrm{decision}}(f(x_i), y_i)}{\sum_{i=1}^{n_{\mathrm{test}}} |y_i^\top w^*(y_i)|}.$$

This metric quantifies the percentage gap in decision quality compared to an oracle with full knowledge. Implementation details can be found in Appendix D.

### 4.1. 2D knapsack Problem

We first studied the 2-dimensional knapsack problem following the setup in (Tang & Khalil, 2024b; Zharmagambetov et al., 2023). The objective is to maximize the value of carried items subject to two distinct capacity constraints (e.g., weight and volume).

This problem is a classic NP-hard combinatorial optimization problem, in which the computation cost grows exponentially with the problem scale. In the experiment, we set feature dimension $p = 5$, number of items $d = 16$, and capacity $W = 20$ for each dimension. Regarding the data generating process, for each trial, the item weight follows $W_{1j}, W_{2j} \overset{\mathrm{i.i.d.}}{\sim} \mathrm{Unif}[3, 8]$ for $j = 1, \ldots, d$. The input features are sampled as $x_i \overset{\mathrm{i.i.d.}}{\sim} \mathcal{N}(0, I_p)$. The item value $y_{ij}$ follows $y_{ij} = f_{ij}^*(x_i)(1 + \epsilon_{ij})$ where $\epsilon_{ij} \overset{\mathrm{i.i.d.}}{\sim} \mathrm{Unif}[-0.5, 0.5]$, and $f_{ij}$ follows:

$$f_{ij}^*(x) = \frac{5}{3.5^{degree}} \left[ \left( \frac{1}{\sqrt{5}}(B^* x_i)_j + 3 \right)^{degree} + 1 \right]$$

where $B^* \in \{0, 1\}^{d \times p}$ has i.i.d. Bernoulli(0.5) entries. The *degree* parameter controls the degree of model misspecification. As *degree* increases, the data-generating process

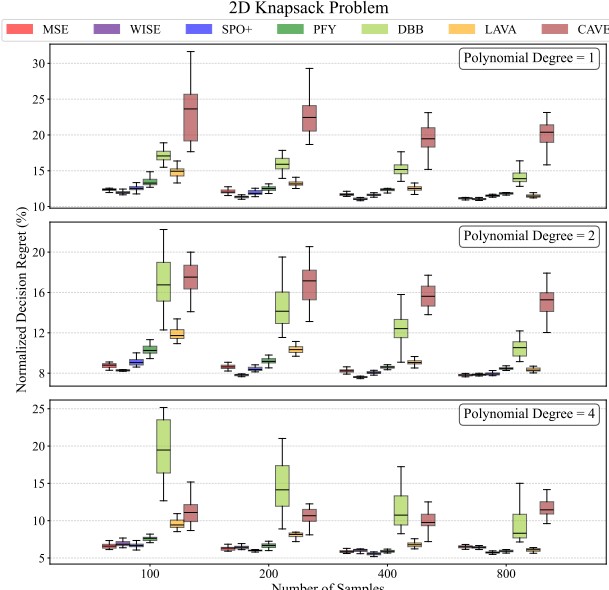

*Figure 3.* Normalized testing set regret vs. number of training samples for 2D knapsack problem. Results are from 20 independent trials with 100 training epochs, under varying degrees of misspecification ($degree = [1, 2, 4]$).

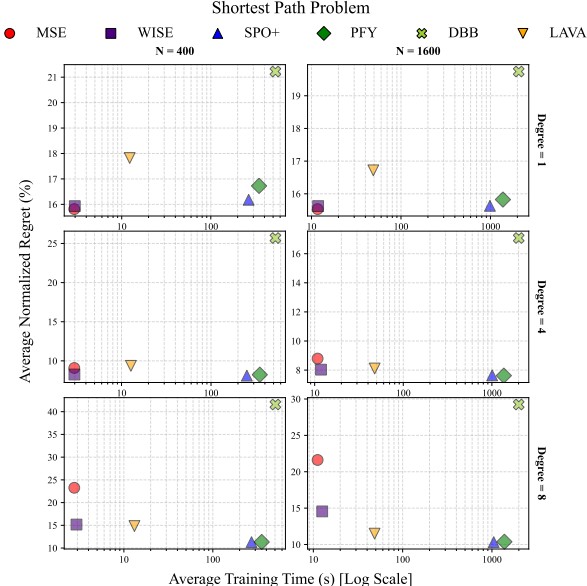

*Figure 4.* Average normalized regret vs. average training time for the shortest path problem. Results are from 20 independent trials with 100 training epochs, under varying degrees of misspecification ($degree = [1, 4, 8]$) and varying number of samples ($N = 400, 1600$).

becomes highly non-linear, making the linear hypothesis class increasingly misspecified.

Figure 3 presents the experimental results. Our method produces high-quality solutions across different degrees of model misspecification and significantly outperforms the benchmarks, including MSE, especially when the training set is small ($n = 100, 200, 400$ and $degree = 1, 2$). Notably, among the DFL benchmarks considered, namely SPO+, PFY, DBB, LAVA, and CAVE, our method is the only solver-free approach in the setting where only true cost vectors are observed rather than past optimal decisions. The proposed WISE loss is highly competitive in terms of both running time and decision regret.

### 4.2. Shortest Path Problem

We adopt the shortest path experimental setup used in (Elmachtoub & Grigas, 2022; Huang & Gupta, 2024; Tang & Khalil, 2024b; Hu et al., 2022; Zharmagambetov et al., 2023). The objective is to minimize travel time on a $5 \times 5$ grid graph ($d = 40$ edges). We evaluate the efficiency-performance trade-off by plotting average test regret against training time.

As shown in Figure 4, our method is significantly faster than DFL benchmarks (4 to 100 times faster). Regarding solution quality, our method outperforms DFL methods under a low misspecification (e.g., degree = 1) and remains competitive under high misspecification. When considering both metrics, our approach offers the most effective end-to-

end learning solution.

It is worth noting that while the $5 \times 5$ shortest path problem allows for efficient polynomial-time solutions, our method still achieves massive efficiency gains. This suggests that our computational advantage will scale significantly for harder, large-scale optimization problems.

### 4.3. Portfolio Optimization

Finally, we adopted the Markowitz model portfolio optimization problem used in (Elmachtoub & Grigas, 2022; Huang & Gupta, 2024; Tang & Khalil, 2024b) where the objective is to maximize the expected portfolio return of 25 stocks subject to a constraint on the variance (risk) of the portfolio, resulting in a convex optimization with a linear objective and quadratic constraints. This experiment provides a close connection to the discussion of strongly convex level sets in Section 3.3, although we featured a mixed setting.

To evaluate robustness against extreme market events, we modify the experimental setup by sampling noise from a Student's $t$-distribution rather than the standard Gaussian noise used in the previous experiments. This setup challenges the learning algorithms to handle heavy-tailed training data.

Figure 5 illustrates the experiment results. Unlike the shortest path and knapsack settings, the performance gaps between methods are smaller. This is expected, as the strongly convex nature of the feasible region ensures a stable map-

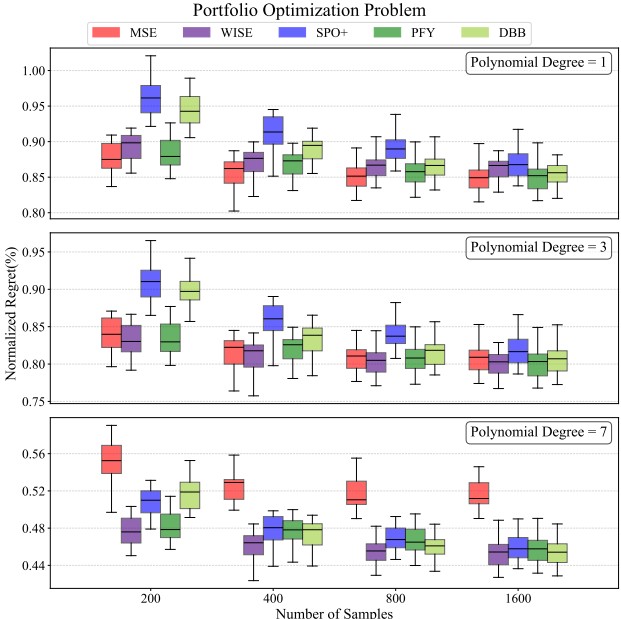

*Figure 5.* Normalized testing set regret vs. number of training samples for portfolio optimization problem. Results are from 20 independent trials with 100 training epochs, under varying degrees of misspecification ($degree = [1, 3, 7]$).

ping from prediction to decision, where small prediction errors translate to proportionally small decision regret. Nevertheless, our method maintains a consistent advantage, particularly in model misspecification regimes.

## 5. Conclusion

We presented a solver-free method that addresses the scalability issue of DFL pipelines. By applying a measure transformation that exploits the underlying geometry of linear optimization, we developed the Weight-Integrated Spherical Error ($\ell_{\text{WISE}}$), which is a surrogate loss that aligns prediction accuracy with decision quality. Theoretically, we show that $\ell_{\text{WISE}}$ is Fisher consistent and provide finite-sample excess risk bounds. Empirically, our method matches or exceeds state-of-the-art decision quality while being significantly faster. We advocate for this measure-transformation-based approach as a blueprint for practical and scalable implementation. Potential future directions include extending the measure transformation idea to capture more distinct problem structures.

## Impact Statement

We propose a scalable DFL method that circumvents the need for computationally expensive optimization solvers during the training loop. This contribution offers broader social impacts in two key areas.

First, our approach promotes environmental sustainability. By making the training process "solver-free", our approach significantly lowers the energy consumption and carbon emissions associated with training complex machine learning models for operations research.

Second, we enhance accessibility and equity in the application of advanced optimization. Traditional DFL methods face scalability issues, often restricting their deployment to large institutions with high-performance computing capabilities. However, by removing the barrier of heavy computational resources, our method allows smaller organizations or entities with limited hardware to deploy a sophisticated DFL pipeline. This could improve efficiency in critical sectors in vehicle routing, public resource allocation, etc.

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

## A. A Formal Proof of the Scale Invariant Property

In this section, we aim to provide a detailed, rigorous proof for $w^*(y) = w^*(\alpha y)$.

Let $\mathcal{S}$ be a non-empty compact set. We first define the optimization solution $w^*(y)$ in two steps: identifying the optimal set and applying a deterministic tie-breaking rule.

Let $W(c)$ denote the set of all optimal solutions for a cost vector $c$:

$$W(c) = \arg\min_{w \in \mathcal{S}} c^\top w = \{w \in \mathcal{S} \mid c^\top w \le c^\top z, \forall z \in \mathcal{S}\}.$$

Let $\tau$ be a deterministic tie-breaking function (selection function) that maps any non-empty compact subset $A \subseteq \mathcal{S}$ to a unique element within that subset:

$$\tau : \mathcal{P}(\mathcal{S}) \setminus \{\emptyset\} \to \mathcal{S} \quad \text{such that} \quad \tau(A) \in A,$$

where $\mathcal{P}(\mathcal{S})$ denote the power set of $\mathcal{S}$. Common examples include choosing the solution with the minimum $\ell_2$ norm, or lexicographically ordering. Note that this rule depends only on the set $A$, not on the vector $y$ that generated it.

The unique solver $w^*(y)$ is defined as the application of this rule to the optimal set:

$$w^*(y) = \tau(W(y)).$$

We first show that the set of candidates $W(y)$ is invariant under positive scaling. Let $\alpha > 0$ and $w \in W(y)$. By definition:

$$y^\top w \le y^\top z, \quad \forall z \in \mathcal{S}.$$

Since $\alpha > 0$, we can multiply the inequality by $\alpha$ without changing its direction:

$$\alpha(y^\top w) \le \alpha(y^\top z) \iff (\alpha y)^\top w \le (\alpha y)^\top z, \quad \forall z \in \mathcal{S}.$$

This condition is equivalent to $w \in W(\alpha y)$. Thus, $W(y) = W(\alpha y)$.

We now evaluate the function $w^*$ at $\alpha y$:

$$\begin{aligned}
w^*(\alpha y) &= \tau(W(\alpha y)) \quad &\text{(By definition of the unique solver)} \\
&= \tau(W(y)) \quad &\text{(Since } W(\alpha y) = W(y)\text{)} \\
&= w^*(y). \quad &\text{(By definition of the unique solver)}
\end{aligned}$$

Therefore, $w^*(y) = w^*(\alpha y)$ for all $\alpha > 0$.

## B. A Discussion of the Measure Transformation Design

This section justifies the necessity of combining Step 1 and Step 2. We argue that neither step is sufficient on its own to produce a Fisher consistent surrogate. This is established through a risk derivation analysis, complemented by a counterexample showing the failure of separated steps.

Let $Q_1$ denote the reweighted measure defined by Step 1. Specifically, we define $Q_1$ that is absolutely continuous with respect to $P$ ($Q_1 \ll P$), and the Radon-Nikodym derivative is defined as:

$$\frac{dQ_1}{dP}(x, y) = \frac{\|y\|}{C}.$$

where $C = \mathbb{E}_P[\|Y\|]$ is a normalizing constant since $\int \|y\| dP(x, y) = \mathbb{E}_P[\|Y\|]$.

Let $Q_2$ denote the projected measure defined by Step 2, i.e., define the mapping $T$ and the push forward measure by

$$T(x, y) = (x, \frac{y}{\|y\|}), \quad Q_2 = P \circ T^{-1}.$$

We first derive the risk expression under $Q_1$:

$$
\begin{aligned}
\mathbb{E}_{(X,Y)\sim Q_1}\left[\ell_{\text{decision}}(f(X),Y)\right] &= \int_{\mathbb{R}^p\times\mathbb{R}^d} \ell_{\text{decision}}(f(x),y)\, dQ_1(x,y) \\
&= \int_{\mathbb{R}^p\times\mathbb{R}^d} \ell_{\text{decision}}(f(x),y)\, \underbrace{\frac{\|y\|}{\mathbb{E}_{(X,Y)\sim P}[\|Y\|]}\, dP(x,y)}_{dQ_1(x,y)} \quad \text{(Radon-Nikodym derivative)} \\
&= \frac{1}{\mathbb{E}_{(X,Y)\sim P}[\|Y\|]} \int_{\mathbb{R}^p\times\mathbb{R}^d} \|y\|\ell_{\text{decision}}(f(x),y)\, dP(x,y).
\end{aligned}
$$

Apparently, the integral term does not equal the decision risk under $P$ for a general distribution. Hence, minimizing over $f$ under $Q_1$ does not yield a Fisher consistent estimator.

Similarly, we consider the risk expression under $Q_2$:

$$
\begin{aligned}
\mathbb{E}_{(X,Z)\sim Q_2}\left[\ell_{\text{decision}}(f(X),Z)\right] &= \int_{\mathbb{R}^p\times\mathbb{S}^{d-1}} \ell_{\text{decision}}(f(x),z)\, dQ_2(x,z) \\
&= \int_{\mathbb{R}^p\times\mathbb{R}^d} \ell_{\text{decision}}(f(x),\frac{y}{\|y\|})\, dP(x,y) \quad \text{(change of measure formula)} \\
&= \int_{\mathbb{R}^p\times\mathbb{R}^d} \frac{1}{\|y\|}\ell_{\text{decision}}(f(x),y)\, dP(x,y). \quad \text{(decision loss scaling property)}
\end{aligned}
$$

Similar to the previous discussion, the integral term also does not equal the decision risk under $P$ for a general distribution, and minimizing over $f$ under $Q_2$ does not yield a Fisher consistent estimator.

After establishing the above expression, we aid it with a simple counterexample. Consider a two-dimensional setting where $X$ is constant and the feasible region is the unit ball, $\mathcal{S} = \{w \in \mathbb{R}^2 : \|w\|_2 \le 1\}$. In this context, minimizing the decision risk is equivalent to estimating the expectation of $Y$ up to a positive scalar.

Let $Y \in \mathbb{R}^2$ take values $(4,0)$ and $(0,1)$, each with probability $\frac{1}{2}$. The true expectation is $\mathbb{E}_P[Y] = (2,\frac{1}{2})$; thus, a Fisher consistent estimator $\hat{Y}$ must satisfy $\hat{Y} = k \cdot (2,\frac{1}{2})$ for some constant $k > 0$.

Now we analyze the estimator obtained under measure $Q_1$ and $Q_2$. Under measure $Q_1$, the probabilities of $Y$ taking values $(4,0)$ and $(0,1)$ become $\frac{4}{5}$ and $\frac{1}{5}$, respectively. Consequently, minimizing Bayes risk under $Q_1$ measure yields $\hat{Y}_1 = (\frac{16}{5},\frac{1}{5})$. Under measure $Q_2$, $Y$ takes value $(1,0)$ and $(0,1)$ both with probability $\frac{1}{2}$. Hence, minimizing Bayes risk under $Q_2$ measure yields $\hat{Y}_2 = (\frac{1}{2},\frac{1}{2})$. Apparently, neither estimator is proportional to the true expectation $\mathbb{E}_P[Y] = (2,\frac{1}{2})$, and hence both are not Fisher consistent. On the other hand, our method yields $\hat{y}_Q = (4/5, 1/5)$, which is in the same direction as the expectation. Figure 6 illustrates this counterexample.

## C. Omitted Proofs

### C.1. Risk Equivalence under Measure Transformation

*Proof of Proposition 2.2.* Recall the definition of $\tilde{Q}$ and $T$ defined in (Step 1)–(Step 2), we have $Q = \tilde{Q} \circ T^{-1}$. By the Change of Variables theorem for push-forward measures $Q$, for the loss $\ell_2(x,z) = \|f(x) - z\|^2$, we have:

$$
\int_{\mathbb{R}^p\times\mathbb{S}^{d-1}} \ell_2(x,z)\, dQ(x,z) = \int_{\mathbb{R}^p\times\mathbb{R}^d} (\ell_2 \circ T)(x,y)\, d\tilde{Q}(x,y).
$$

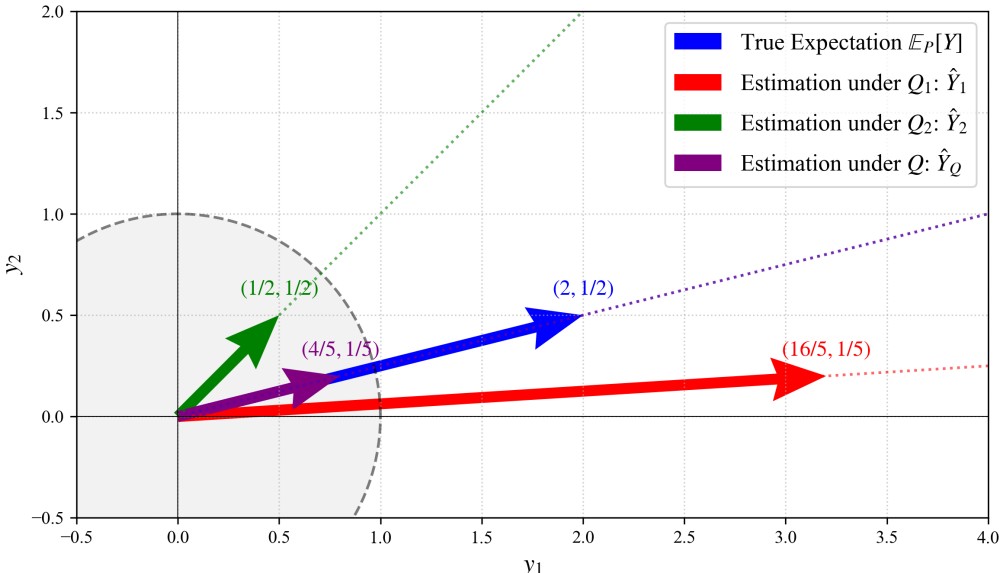

*Figure 6.* Visual illustration of the counterexample. The shaded area represents the feasible region $\mathcal{S} = \{w \in \mathbb{R}^2 : \|w\|_2 \le 1\}$. The blue arrow denotes the true expectation $\mathbb{E}_P[Y]$. The red and green arrows correspond to estimators $\hat{Y}_1$ and $\hat{Y}_2$ derived under measures $Q_1$ and $Q_2$, respectively, while the purple arrow represents the estimator $\hat{Y}_Q$ from our proposed method. Fisher consistency requires the estimator to be collinear with the true expectation. As shown, $\hat{Y}_1$ and $\hat{Y}_2$ deviate from the direction of $\mathbb{E}_P[Y]$, whereas our estimator $\hat{Y}_Q$ maintains the correct direction.

Substituting the definition of $\tilde{Q}$ via the Radon-Nikodym derivative:

$$
\begin{aligned}
\mathbb{E}_{(X,Z)\sim Q}\left[\|f(X) - Z\|^2\right] &= \int_{\mathbb{R}^p \times \mathbb{S}^{d-1}} \|f(x) - z\|^2 dQ(x,z) \\
&= \int_{\mathbb{R}^p \times \mathbb{R}^d} \|f(x) - \frac{y}{\|y\|}\|^2 d\tilde{Q}(x,y) \quad \text{(change of measure formula)} \\
&= \int_{\mathbb{R}^p \times \mathbb{R}^d} \left\|f(x) - \frac{y}{\|y\|}\right\|^2 \underbrace{\frac{\|y\|}{\mathbb{E}_{X,Y\sim P}[\|Y\|]}}_{d\tilde{Q}(x,y)} dP(x,y) \quad \text{(Radon-Nikodym derivative)} \\
&= \frac{1}{\mathbb{E}_{(X,Y)\sim P}[\|Y\|]} \int_{\mathbb{R}^p \times \mathbb{R}^d} \|y\| \cdot \left\|f(x) - \frac{y}{\|y\|}\right\|^2 dP(x,y) \\
&= \frac{1}{\mathbb{E}_{(X,Y)\sim P}[\|Y\|]} \mathbb{E}_{(X,Y)\sim P}\left[\|Y\| \cdot \left\|f(X) - \frac{Y}{\|Y\|}\right\|^2\right].
\end{aligned}
$$

Hence, the equivalence holds. □

*Proof of Proposition 2.1.* By the Change of Variables theorem for push-forward measures $Q$, for the loss $\ell_{\text{decision}}(x, z)$, we have:

$$
\int_{\mathbb{R}^p \times \mathbb{S}^{d-1}} \ell_{\text{decision}}(x, z)\, dQ(x, z) = \int_{\mathbb{R}^p \times \mathbb{R}^d} (\ell_{\text{decision}} \circ T)(x, y)\, d\tilde{Q}(x, y).
$$

Substituting the definition of $\tilde{Q}$ via the Radon-Nikodym derivative:

$$
\begin{aligned}
\mathbb{E}_{(X,Z)\sim Q}\left[\ell_{\text{decision}}(f(X),Z)\right] &= \int_{\mathbb{R}^p\times\mathbb{S}^{d-1}} \ell_{\text{decision}}(f(x),z)\,dQ(x,z) \\
&= \int_{\mathbb{R}^p\times\mathbb{R}^d} \ell_{\text{decision}}\left(f(x),\frac{y}{\|y\|}\right) d\tilde{Q}(x,y) \\
&= \int_{\mathbb{R}^p\times\mathbb{R}^d} \frac{1}{\|y\|}\ell_{\text{decision}}(f(x),y)\,d\tilde{Q}(x,y) \quad \text{(decision loss scaling property)} \\
&= \int_{\mathbb{R}^p\times\mathbb{R}^d} \frac{1}{\|y\|}\ell_{\text{decision}}(f(x),y)\underbrace{\frac{\|y\|}{\mathbb{E}_{(X,Y)\sim P}[\|Y\|]}\,dP(x,y)}_{d\tilde{Q}(x,y)} \\
&= \frac{1}{\mathbb{E}_{(X,Y)\sim P}[\|Y\|]}\int_{\mathbb{R}^p\times\mathbb{R}^d} \ell_{\text{decision}}(f(x),y)\,dP(x,y) \\
&= \frac{1}{\mathbb{E}_{(X,Y)\sim P}[\|Y\|]}\mathbb{E}_{(X,Y)\sim P}\left[\ell_{\text{decision}}(f(X),Y)\right].
\end{aligned}
$$

Hence the equivalence holds. $\qquad\square$

## C.2. Analytical Properties and Fisher Consistency

*Proof of Proposition 3.2.* First, we prove the smoothness property. A function $g$ is $\beta$-smooth if its gradient is $\beta$-Lipschitz, which is implied if the spectral norm of its Hessian is bounded by $\beta$. Recall the definition of $\ell_{\text{WISE}}$, we have

$$
\ell_{\text{WISE}}(\hat{y},y) = \|y\|\left\|\hat{y}-\frac{y}{\|y\|}\right\|^2.
$$

Taking the gradient of $\ell_{\text{WISE}}$ with respect to $\hat{y}$:

$$
\nabla_{\hat{y}}\ell_{\text{WISE}}(\hat{y},y) = 2\|y\|\hat{y}-2y.
$$

The Hessian with respect to $\hat{y}$ is:

$$
\nabla_{\hat{y}}^2\ell_{\text{WISE}}(\hat{y},y) = 2\|y\|I_d,
$$

where $I_d$ is the identity matrix in $\mathbb{R}^d$. The eigenvalues of the Hessian are uniformly equal to $2\|y\|$. Since $\|y\|\leq B_y$ by assumption, the maximum eigenvalue is bounded by $2B_y$. Thus, $\ell_{\text{WISE}}$ is $2B_y$-smooth.

Then, we prove the local Lipschitz property. A differentiable function is $L$-Lipschitz over a convex domain if the norm of its gradient is bounded by $L$ everywhere in that domain. Consider the norm of the gradient derived above:

$$
\begin{aligned}
\|\nabla_{\hat{y}}\ell_{\text{WISE}}(\hat{y},y)\| &= \|2\|y\|\hat{y}-2y\| \\
&\leq 2\|\|y\|\hat{y}\|+2\|y\| \quad \text{(Triangle Inequality)} \\
&= 2\|y\|\|\hat{y}\|+2\|y\| \\
&= 2\|y\|(\|\hat{y}\|+1).
\end{aligned}
$$

Using the boundedness assumptions $\|\hat{y}\|\leq B$ and $\|y\|\leq B_y$, we obtain:

$$
\|\nabla_{\hat{y}}\ell_{\text{WISE}}(\hat{y},y)\| \leq 2B_y(B+1).
$$

Therefore, $\ell_{\text{WISE}}$ is locally Lipschitz continuous with constant $L=2B_y(B+1)$.

We further prove the boundedness property. We bound the loss function value itself. Using the definition of $\ell_{\text{WISE}}$ and the Triangle Inequality on the norm term:

$$
\begin{aligned}
\ell_{\text{WISE}}(\hat{y},y) &= \|y\|\left\|\hat{y}-\frac{y}{\|y\|}\right\|^2 \\
&\leq B_y\left(\|\hat{y}\|+\left\|\frac{y}{\|y\|}\right\|\right)^2 \\
&= B_y(\|\hat{y}\|+1)^2.
\end{aligned}
$$

Given $\|\hat{y}\| \leq B$, we have:

$$\ell_{\text{WISE}}(\hat{y}, y) \leq B_y (B+1)^2.$$

This completes the proof. □

*Proof of Proposition 3.3.* Let $f$ denote a measurable function from $\mathbb{R}^p \to \mathbb{R}^d$. Let $P$ denote the joint distribution of $(X, Y)$. By the Tower property of conditional expectation, we can express the risk in terms of the conditional expected loss:

$$R_P^{\ell_{\text{WISE}}}(f) = \mathbb{E}_X \left[ \mathbb{E}[\ell_{\text{WISE}}(f(X), Y) \mid x] \right].$$

We analyze the inner conditional expectation. Recall that the wise loss is defined as $\ell_{\text{WISE}}(f(X), Y) = \|Y\| \|f(X) - \frac{Y}{\|Y\|}\|^2$. Expanding this definition with respect to the conditional expectation:

$$\mathbb{E}[\ell_{\text{WISE}}(f(x), Y) \mid x] = \mathbb{E}\left[ \|Y\| \left\| f(X) - \frac{Y}{\|Y\|} \right\|^2 \Big| x \right]$$

$$= \mathbb{E}\left[ \|Y\| \|f(x)\|^2 - 2Y^\top f(x) + \|Y\| \mid x \right]$$

$$= \|f(x)\|^2 \mathbb{E}[\|Y\| \mid x] - 2f(x)^\top \mathbb{E}[Y \mid x] + \mathbb{E}[\|Y\| \mid x].$$

We aim to complete the square with respect to the predictor $f(x)$, i.e., want to show $\mathbb{E}[\ell_{\text{WISE}}(f(x), Y) \mid x]$ can be expressed as

$$\mathbb{E}[\ell_{\text{WISE}}(f(x), Y) \mid x] = \mathbb{E}[\|Y\| \mid x] \left\| f(x) - \frac{\mathbb{E}[Y \mid x]}{\mathbb{E}[\|Y\| \mid x]} \right\|^2 + C(x),$$

where $C(x)$ is a constant term independent of $f(x)$.

To show this, we first identify the target $\tau(x) = \frac{\mathbb{E}[Y|x]}{\mathbb{E}[\|Y\| |x]}$. Provided that $\mathbb{E}[\|Y\| \mid x] > 0$, we can rewrite the variable terms:

$$\mathbb{E}[\|Y\| \mid x] \|f(x) - \tau(x)\|^2 = \mathbb{E}[\|Y\| \mid x] \left( \|f(x)\|^2 - 2f(x)^\top \tau(x) + \|\tau(x)\|^2 \right)$$

$$= \|f(x)\|^2 \mathbb{E}[\|Y\| \mid x] - 2f(x)^\top \mathbb{E}[Y \mid x] + \|\tau(x)\|^2 \mathbb{E}[\|Y\| \mid x]$$

which matched our target with $C(x) = \mathbb{E}[\|Y\| \mid x](1 - \|\tau(x)\|^2)$.

The total risk is then:

$$R_P^{\ell_{\text{WISE}}}(f) = \mathbb{E}_X \left[ \mathbb{E}[\|Y\| \mid x] \left\| f(x) - \frac{\mathbb{E}[Y \mid x]}{\mathbb{E}[\|Y\| \mid x]} \right\|^2 + C(x) \right].$$

Since $\mathbb{E}[\|Y\| \mid x] > 0$ almost everywhere with respect to $P$ and the squared norm is non-negative, the expectation is minimized if and only if the term inside the norm is zero almost everywhere. Therefore, the Bayes risk minimizer $f^*$ is unique almost everywhere and satisfies:

$$f^*(x) = \frac{\mathbb{E}[Y \mid x]}{\mathbb{E}[\|Y\| \mid x]} \quad \text{a.s.}$$

□

*Proof of Theorem 3.4.* From Proposition 3.3, we know that the Bayes risk minimizer of $\ell_{\text{WISE}}$ satisfies $f(x) = \frac{\mathbb{E}[y|x]}{\mathbb{E}[\|y\| |x]}$.

This minimizer is a strictly positive scalar multiple of the conditional expectation $\mathbb{E}[y \mid x]$. By the Scale Invariance property of the optimization oracle $w^*$, scaling the input vector by a positive constant does not change the optimal decision:

$$w^*(f(x)) = w^* \left( \frac{\mathbb{E}[y \mid x]}{\mathbb{E}[\|y\| \mid x]} \right) = w^*(\mathbb{E}[y \mid x]).$$

Since $w^*(\mathbb{E}[y \mid x])$ minimizes the expected cost $\mathbb{E}[y \mid x]^\top w$, $f$ minimizes the decision regret. □

### C.3. Calibration Risk Bound

To analyze the calibration of $\ell_{\text{WISE}}$, we first prove a supplement lemma.

**Lemma C.1** (Equivalent Form of $\ell_{\text{WISE}}$ Excess Regret)**.** *The excess regret of $\ell_{\text{WISE}}$ admits the following equivalent form:*

$$R_P^{\ell_{\text{WISE}}}(f) - R_P^{\ell_{\text{WISE}}}(f^*) = \mathbb{E}_P\left[\mathbb{E}[\|Y\| \mid X] \cdot \|f(X) - f^*(X)\|^2\right],$$

*where $f^*(x) = \frac{\mathbb{E}_P[Y|x]}{\mathbb{E}_P[\|Y\||x]}$ is the Bayes risk minimizer of $\ell_{\text{WISE}}$.*

*Proof of Lemma C.1.* We begin by expanding the definition of the excess regret. Using the definition of $R_P^{\ell_{\text{WISE}}}$, we have:

$$R_P^{\ell_{\text{WISE}}}(f) - R_P^{\ell_{\text{WISE}}}(f^*) = \mathbb{E}_P\left[\|Y\|\left\|f(X) - \frac{Y}{\|Y\|}\right\|^2\right] - \mathbb{E}_P\left[\|Y\|\left\|f^*(X) - \frac{Y}{\|Y\|}\right\|^2\right].$$

Next, we expand the squared norms. Note that the terms involving only $Y$ (specifically $\|Y\| \cdot \|Y/\|Y\|\|^2$) cancel out, and we use the linearity of expectation to group the remaining terms:

$$R_P^{\ell_{\text{WISE}}}(f) - R_P^{\ell_{\text{WISE}}}(f^*) = \mathbb{E}_P\left[\|Y\|(\|f(X)\|^2 - \|f^*(X)\|^2) - 2Y^\top(f(X) - f^*(X))\right].$$

By the tower property of conditional expectation, we get:

$$= \mathbb{E}_P\left[\mathbb{E}_P[\|Y\| \mid X](\|f(X)\|^2 - \|f^*(X)\|^2) - 2\mathbb{E}_P[Y \mid X]^\top(f(X) - f^*(X))\right].$$

Substituting the explicit expression for the Bayes risk minimizer, $f^*(X) = \frac{\mathbb{E}_P[Y|X]}{\mathbb{E}_P[\|Y\||X]}$, into the equation yields:

$$= \mathbb{E}_P\left[\mathbb{E}_P[\|Y\| \mid X]\left(\|f(X)\|^2 - \left\|\frac{\mathbb{E}_P[Y \mid X]}{\mathbb{E}_P[\|Y\| \mid X]}\right\|^2\right) - 2\mathbb{E}_P[Y \mid X]^\top\left(f(X) - \frac{\mathbb{E}_P[Y \mid X]}{\mathbb{E}_P[\|Y\| \mid X]}\right)\right].$$

Expanding the inner product and simplifying the scalar terms allows us to regroup the expression. We distribute the $\mathbb{E}_P[\|Y\| \mid X]$ term and rearrange to prepare for completing the square:

$$= \mathbb{E}_P\left[\mathbb{E}[\|Y\| \mid X]\|f(X)\|^2 - 2\mathbb{E}_P[Y \mid X]^\top f(X) + 2\frac{\|\mathbb{E}_P[Y \mid X]\|^2}{\mathbb{E}_P[\|Y\| \mid X]} - \mathbb{E}_P[\|Y\| \mid X]\left\|\frac{\mathbb{E}_P[Y \mid X]}{\mathbb{E}_P[\|Y\| \mid X]}\right\|^2\right]$$

$$= \mathbb{E}_P\left[\mathbb{E}_P[\|Y\| \mid X]\|f(X)\|^2 - 2\mathbb{E}_P[\|Y\| \mid X]\frac{\mathbb{E}_P[Y \mid X]^\top}{\mathbb{E}_P[\|Y\| \mid X]}f(X) + \mathbb{E}_P[\|Y\| \mid X]\left\|\frac{\mathbb{E}_P[Y \mid X]}{\mathbb{E}_P[\|Y\| \mid X]}\right\|^2\right].$$

The term inside the expectation is now:

$$= \mathbb{E}_P\left[\mathbb{E}_P[\|Y\| \mid X]\left\|f(X) - \frac{\mathbb{E}_P[Y \mid X]}{\mathbb{E}_P[\|Y\| \mid X]}\right\|^2\right].$$

Finally, reintroducing the definition of $f^*(X)$ concludes the proof:

$$= \mathbb{E}_P\left[\mathbb{E}_P[\|Y\| \mid X]\|f(X) - f^*(X)\|^2\right].$$

$\square$

Equipped with the above lemma, we are ready to analyze the calibration bound.

*Proof of Proposition 3.5.* By the previous definition, let $f^*(x) = \frac{\mathbb{E}_P[Y|X=x]}{\mathbb{E}_P[\|Y\| \mid X=x]}$ denote the Bayes risk minimizer of $\ell_{\text{WISE}}$. We begin by expressing the excess decision risk using the tower property of conditional expectation:

$$R_P^{\ell_{\text{decision}}}(f) - R_P^* = \mathbb{E}_P\left[Y^\top w^*(f(X)) - Y^\top w^*(\mathbb{E}_P[Y|X])\right]$$
$$= \mathbb{E}_P\left[\mathbb{E}_P[Y \mid X]^\top (w^*(f(X)) - w^*(\mathbb{E}_P[Y|X]))\right].$$

Next, we introduce the normalizing term $\mathbb{E}_P[\|Y\| \mid X]$ to substitute the Bayes risk minimizer $f^*(X)$. Note that $w^*(\mathbb{E}_P[Y|x]) = w^*(f^*(x))$ holds since $\mathbb{E}_P[Y|x]$ and $f^*$ only differs in a constant factor:

$$= \mathbb{E}_P\left[\mathbb{E}_P[\|Y\| \mid X]\frac{\mathbb{E}_P[Y \mid X]^\top}{\mathbb{E}_P[\|Y\| \mid X]}(w^*(f(X)) - w^*(\mathbb{E}_P[Y|X]))\right]$$
$$= \mathbb{E}_P\left[\mathbb{E}_P[\|Y\| \mid X]f^*(X)^\top (w^*(f(X)) - w^*(f^*(X)))\right].$$

We now add and subtract the term $f(X)^\top w^*(f(X))$ inside the expectation to utilize the optimality properties of the optimization oracle $w^*(\cdot)$. Specifically, we use the property that $w^*(f(X))$ optimizes the inner product with $f(X)$, which implies $f(X)^\top w^*(f(X)) \leq f(X)^\top w^*(f^*(X))$:

$$= \mathbb{E}_P\left[\mathbb{E}_P[\|Y\| \mid X](f^*(X)^\top w^*(f(X)) - f(X)^\top w^*(f(X)) + f(X)^\top w^*(f(X)) - f^*(X)^\top w^*(f^*(X)))\right]$$
$$\leq \mathbb{E}_P\left[\mathbb{E}_P[\|Y\| \mid X](f^*(X)^\top w^*(f(X)) - f(X)^\top w^*(f(X)) + f(X)^\top w^*(f^*(X)) - f^*(X)^\top w^*(f^*(X)))\right].$$

Regrouping the terms by the decision vectors $w^*(f(X))$ and $w^*(f^*(X))$ reveals the structure for applying the Cauchy-Schwarz inequality:

$$= \mathbb{E}_P\left[\mathbb{E}_P[\|Y\| \mid X](w^*(f(X))^\top (f^*(X) - f(X)) + w^*(f^*(X))^\top (f(X) - f^*(X)))\right]$$
$$\leq \mathbb{E}_P\left[\mathbb{E}_P[\|Y\| \mid X](\|w^*(f(X))\|\|f^*(X) - f(X)\| + \|w^*(f^*(X))\|\|f(X) - f^*(X)\|)\right].$$

Using the boundedness of the feasible region $\mathcal{S}$, where $\|w\| \leq B_{\mathcal{S}}$ for all $w \in \mathcal{S}$, we simplify the expression:

$$\leq 2B_{\mathcal{S}}\mathbb{E}_P\left[\mathbb{E}_P[\|Y\| \mid X]\|f^*(X) - f(X)\|\right].$$

Finally, we apply Jensen's inequality ($\mathbb{E}[Z] \leq \sqrt{\mathbb{E}[Z^2]}$) and the boundedness of the label norms ($\mathbb{E}_P[\|Y\| \mid X] \leq B_y$) to relate this bound back to the pairwise excess regret established in Lemma C.1:

$$\leq 2B_{\mathcal{S}}\sqrt{\mathbb{E}_P\left[\mathbb{E}_P[\|Y\| \mid X]^2\|f^*(X) - f(X)\|^2\right]}$$
$$\leq 2B_{\mathcal{S}}B_y^{\frac{1}{2}}\sqrt{\mathbb{E}_P\left[\mathbb{E}_P[\|Y\| \mid X]\|f^*(X) - f(X)\|^2\right]}$$
$$= 2B_{\mathcal{S}}B_y^{\frac{1}{2}}\sqrt{R_P^{\ell_{\text{WISE}}}(f) - R_P^{\ell_{\text{WISE}}}(f^*)}.$$

$\square$

### C.4. Excess Regret Bound

In this section, we provide the proof for Theorem 3.7. The high level idea is to prove the strong central condition (Def. C.3) via the Bernstein type condition under the squared loss. This Bernstein type bound is further ensured by the star-shaped condition of the hypothesis class. Assume $\|Y\| > 0$ almost surely. Define the deterministic maps

$$s(y) := \sqrt{\|y\|}, \qquad v(y) := \frac{y}{\sqrt{\|y\|}}.$$

For any $f : \mathbb{R}^p \to \mathbb{R}^d$, define the associated function

$$g_f : \mathbb{R}^p \times \mathbb{R}^d \to \mathbb{R}^d, \qquad g_f(x, y) := s(y) f(x) = \sqrt{\|y\|} f(x). \tag{1}$$

Let the induced hypothesis class be

$$\mathcal{G} := \{g_f : f \in \mathcal{F}\}.$$

Define the squared loss on $(x, y) \in \mathbb{R}^p \times \mathbb{R}^d$ by

$$\ell_{\text{sq}}(g; x, y) := \left\| g(x, y) - v(y) \right\|^2 = \left\| g(x, y) - \frac{y}{\sqrt{\|y\|}} \right\|^2.$$

**Proposition C.2** (WISE is a squared loss under $P$). *For any measurable $f : \mathbb{R}^p \to \mathbb{R}^d$ and the induced $g_f$ in (1),the following hold*

$$\ell_{\text{WISE}}(f(x), y) = \ell_{\text{sq}}(g_f; x, y), \qquad \forall(x, y) : \|y\| > 0, \tag{2}$$

$$R_P^{\ell_{\text{WISE}}}(f) := \mathbb{E}_{(X,Y) \sim P}[\ell_{\text{WISE}}(f(X), Y)] = \mathbb{E}_{(X,Y) \sim P}[\ell_{\text{sq}}(g_f; X, Y)] =: R_P^{\text{sq}}(g_f), \tag{3}$$

$$\frac{1}{n} \sum_{i=1}^{n} \ell_{\text{WISE}}(f(X_i), Y_i) = \frac{1}{n} \sum_{i=1}^{n} \ell_{\text{sq}}(g_f; X_i, Y_i). \tag{4}$$

*Proof.* Fix $(x, y)$ with $\|y\| > 0$. Then

$$\ell_{\text{sq}}(g_f; x, y) = \left\| \sqrt{\|y\|} f(x) - \frac{y}{\sqrt{\|y\|}} \right\|^2 = \left\| \sqrt{\|y\|} \left( f(x) - \frac{y}{\|y\|} \right) \right\|^2$$

$$= \|y\| \left\| f(x) - \frac{y}{\|y\|} \right\|^2 = \ell_{\text{WISE}}(f(x), y),$$

which proves (2). Taking expectation gives (3), and summing over samples gives (4). $\qquad\square$

We define the (strong) $\eta$-central condition (a.k.a. exponential stochastic mixability) as follows.

**Definition C.3** (Strong $\eta$-central condition). We say the learning problem $(P, \ell, \mathcal{H})$ satisfies the strong $\eta$-central condition (with comparator $h^* \in \mathcal{H}$) if there exists $\eta > 0$ and $h^* \in \arg\min_{h \in \mathcal{H}} \mathbb{E}[\ell(h; Z)]$ such that for all $h \in \mathcal{H}$,

$$\mathbb{E}\left[ \exp\left( -\eta\left( \ell(h; Z) - \ell(h^*; Z) \right) \right) \right] \leq 1.$$

For bounded losses, the (strong) central condition is essentially equivalent to (generalized) Bernstein conditions; see Theorem 5.4 in (van Erven et al., 2015).

**Definition C.4** ($(1, B_0)$-Bernstein condition). We say $(P, \ell, \mathcal{H})$ satisfies the $(1, B_0)$-Bernstein condition (with comparator $h^*$) if there exists $B_0 > 0$ and $h^* \in \arg\min_{h \in \mathcal{H}} \mathbb{E}[\ell(h; Z)]$ such that for all $h \in \mathcal{H}$,

$$\mathbb{E}\left[ \left( \ell(h; Z) - \ell(h^*; Z) \right)^2 \right] \leq B_0 \cdot \mathbb{E}[\ell(h; Z) - \ell(h^*; Z)]. \tag{5}$$

The next lemma shows that for squared loss, star-shapedness at the risk minimizer (and boundedness) yields a Bernstein condition automatically.

**Lemma C.5** (Bernstein for bounded squared loss under star-shapedness). *Let $Z \sim P$, let $v : \mathcal{Z} \to \mathbb{R}^d$ be measurable, and define $\ell_{\text{sq}}(g; Z) := \|g(Z) - v(Z)\|^2$ for measurable $g : \mathcal{Z} \to \mathbb{R}^d$. Assume:*

*(a) (**Star-shapedness at $g^*$**) $\mathcal{G}$ is star-shaped at $g^* \in \arg\min_{g \in \mathcal{G}} \mathbb{E}[\ell_{\text{sq}}(g; Z)]$, i.e. $(1 - \lambda)g^* + \lambda g \in \mathcal{G}$ for all $g \in \mathcal{G}$ and $\lambda \in [0, 1]$;*

*(b) (**Bounded range**) there exist $G_{\max}, V_{\max} > 0$ such that $\|g(Z)\| \leq G_{\max}$ for all $g \in \mathcal{G}$ a.s. and $\|v(Z)\| \leq V_{\max}$ a.s.*

*Then $(P, \ell_{\text{sq}}, \mathcal{G})$ satisfies the $(1, B_0)$-Bernstein condition with comparator $g^*$ and constant*

$$B_0 = 4 \left( G_{\max} + V_{\max} \right)^2. \tag{6}$$

*Proof.* Fix $g \in \mathcal{G}$ and define $g_\lambda := (1 - \lambda)g^* + \lambda g$ for $\lambda \in [0, 1]$. By star-shapedness, $g_\lambda \in \mathcal{G}$. Since $g^*$ minimizes the squared-risk $R(g) := \mathbb{E}\|g(Z) - v(Z)\|^2$, we have $R(g_\lambda) - R(g^*) \geq 0$ for all $\lambda \in [0, 1]$. Expanding,

$$R(g_\lambda) - R(g^*) = \lambda^2 \, \mathbb{E}\|g - g^*\|^2 + 2\lambda \, \mathbb{E}\langle g - g^*, \, g^* - v\rangle.$$

Divide by $\lambda > 0$ and let $\lambda \downarrow 0$ to obtain $\mathbb{E}\langle g - g^*, \, g^* - v\rangle \geq 0$. Hence,

$$\mathbb{E}[\ell_{\mathrm{sq}}(g; Z) - \ell_{\mathrm{sq}}(g^*; Z)] = R(g) - R(g^*) = \mathbb{E}\|g - g^*\|^2 + 2\mathbb{E}\langle g - g^*, \, g^* - v\rangle \; \geq \; \mathbb{E}\|g - g^*\|^2.$$

Now, pointwise,

$$\begin{aligned}
\left|\ell_{\mathrm{sq}}(g; Z) - \ell_{\mathrm{sq}}(g^*; Z)\right| &= \left|\|g - v\|^2 - \|g^* - v\|^2\right| \\
&= \left|\langle g - g^*, \, g + g^* - 2v\rangle\right| \leq \|g - g^*\| \, (\|g\| + \|g^*\| + 2\|v\|).
\end{aligned}$$

Using $\|g\|, \|g^*\| \leq G_{\max}$ and $\|v\| \leq V_{\max}$ a.s., we get

$$\left|\ell_{\mathrm{sq}}(g; Z) - \ell_{\mathrm{sq}}(g^*; Z)\right| \leq 2(G_{\max} + V_{\max}) \, \|g - g^*\|.$$

Squaring and taking expectation yields

$$\mathbb{E}\left[\left(\ell_{\mathrm{sq}}(g; Z) - \ell_{\mathrm{sq}}(g^*; Z)\right)^2\right] \leq 4(G_{\max} + V_{\max})^2 \, \mathbb{E}\|g - g^*\|^2 \leq 4(G_{\max} + V_{\max})^2 \, \mathbb{E}[\ell_{\mathrm{sq}}(g; Z) - \ell_{\mathrm{sq}}(g^*; Z)],$$

which is (5) with $B_0$ given by (6). $\qquad\square$

Under Assumption 3.1, we have

$$\|v(Y)\| = \sqrt{\|Y\|} \leq \sqrt{B_y}, \qquad \|g_f(X, Y)\| = \sqrt{\|Y\|} \, \|f(X)\| \leq \sqrt{B_y} \, B,$$

so in Lemma C.5 we may take

$$V_{\max} = \sqrt{B_y}, \qquad G_{\max} = \sqrt{B_y} \, B, \qquad \Rightarrow \qquad B_0 = 4B_y(B + 1)^2. \tag{7}$$

Moreover, Assumption 3.1 implies the squared loss is bounded:

$$0 \leq \ell_{\mathrm{sq}}(g; X, Y) = \|g(X, Y) - v(Y)\|^2 \leq (G_{\max} + V_{\max})^2 = B_y(B + 1)^2 \quad \text{a.s.} \tag{8}$$

Finally, if $\mathcal{F}$ is star-shaped at $f^*$, then $\mathcal{G}$ is star-shaped at $g^*$ since $g_{(1-\lambda)f + \lambda f^*} = (1 - \lambda)g_f + \lambda g_{f^*}$.

**Lemma C.6** (VC-linear-subgraph dimension transfer). *Let* VC-LS$(\cdot)$ *denote the VC-linear-subgraph dimension. Then*

$$\mathrm{VC\text{-}LS}(\mathcal{G}) \; \leq \; \mathrm{VC\text{-}LS}(\mathcal{F}).$$

*In particular, if* VC-LS$(\mathcal{F}) \leq \nu$*, then* VC-LS$(\mathcal{G}) \leq \nu$*.*

*Proof.* Consider the VC-linear-subgraph sets for $\mathcal{G}$:

$$\mathcal{G}^\circ := \left\{ (u, \beta, t) : \beta^\top g(u) \leq t \right\}_{g \in \mathcal{G}}, \qquad u = (x, y), \; \beta \in \mathbb{R}^d, \; t \in \mathbb{R}.$$

For any $g = g_f \in \mathcal{G}$ and any $u = (x, y)$ with $\|y\| > 0$,

$$\beta^\top g_f(u) \leq t \quad \Longleftrightarrow \quad \beta^\top \left(\sqrt{\|y\|} f(x)\right) \leq t \quad \Longleftrightarrow \quad \beta^\top f(x) \leq \frac{t}{\sqrt{\|y\|}}.$$

Thus, on any fixed collection of points $u_i = (x_i, y_i)$, the dichotomies induced by $\mathcal{G}^\circ$ correspond to dichotomies induced by $\mathcal{F}^\circ$ evaluated at $(x_i, \beta_i, t_i/\sqrt{\|y_i\|})$. Therefore $\mathcal{G}^\circ$ cannot shatter more points than $\mathcal{F}^\circ$, and VC-LS$(\mathcal{G}) \leq$ VC-LS$(\mathcal{F})$. $\quad\square$

*Proof of Theorem 3.7.* By Proposition C.2, for any $f \in \mathcal{F}$ with induced $g_f \in \mathcal{G}$,

$$R_P^{\ell_{\text{WISE}}}(f) = R_P^{\text{sq}}(g_f).$$

In particular, defining $\hat{g}_n := g_{\hat{f}_n}$ and $g^* := g_{f^*}$, we have the *exact* excess-risk identity

$$R_P^{\ell_{\text{WISE}}}(\hat{f}_n) - R_P^{\ell_{\text{WISE}}}(f^*) = R_P^{\text{sq}}(\hat{g}_n) - R_P^{\text{sq}}(g^*). \tag{9}$$

Under Assumption 3.1, the squared loss is bounded as in (8). Moreover, the star-shaped assumption on $\mathcal{F}$ implies $\mathcal{G}$ is star-shaped at $g^*$, and thus by Lemma C.5 (with the specialization (7)), the squared-loss problem $(P, \ell_{\text{sq}}, \mathcal{G})$ satisfies the $(1, B_0)$-Bernstein condition with $B_0 = 4B_y(B+1)^2$. Since the loss is bounded, Theorem 5.4 of (van Erven et al., 2015) implies that $(P, \ell_{\text{sq}}, \mathcal{G})$ satisfies a (strong) $\eta$-central condition for some $\eta > 0$ depending only on $B_0$ (and hence only on $B, B_y$).

By (4), $\hat{g}_n$ is an ERM over $\mathcal{G}$ for the bounded loss $\ell_{\text{sq}}$. By Lemma C.6, $\ell_{\text{sq}} \circ \mathcal{G}$ has VC-type metric entropy. Therefore, we may apply Theorem 7.7 of (van Erven et al., 2015) to obtain that with probability at least $1 - \delta$,

$$R_P^{\text{sq}}(\hat{g}_n) - R_P^{\text{sq}}(g^*) \ \leq \ \tilde{C}_2 \cdot \frac{\nu \log(C_1 n) + \log(1/\delta)}{n}, \tag{10}$$

for constants $C_1, \tilde{C}_2 > 0$ depending only on $(B, B_y)$ (through the bounded range and the implied $\eta$).

Combining (9) and (10), we have

$$R_P^{\ell_{\text{WISE}}}(\hat{f}_n) - R_P^{\ell_{\text{WISE}}}(f^*) \ \leq \ \tilde{C}_2 \cdot \frac{\nu \log(C_1 n) + \log(1/\delta)}{n}. \tag{11}$$

Finally, applying Proposition 3.5 yields

$$R_P^{\ell_{\text{decision}}}(\hat{f}_n) - R_P^* \ \leq \ 2B_{\mathcal{S}} B_y^{\frac{1}{2}} \sqrt{R_P^{\ell_{\text{WISE}}}(\hat{f}_n) - R_P^{\ell_{\text{WISE}}}(f^*)} \ \leq \ C_2 \sqrt{\frac{\nu \log(C_1 n) + \log(1/\delta)}{n}},$$

where $C_1, C_2$ are universal constants only depending on $B, B_y, B_{\mathcal{S}}$ □

## C.5. Calibration and Excess Regret Bound for Strongly Convex Level Sets

We first recall Lemma 4.1 and Lemma 4.2 from (Liu & Grigas, 2021).

**Lemma C.7.** *(Liu & Grigas, 2021)* *Suppose that Assumption 3.8 holds. Then, for any $c_1, c_2 \in \mathbb{R}^d$, it holds that*

$$c_1^\top (w^*(c_2) - w^*(c_1)) \leq \frac{L}{2\sqrt{2\mu(r - g_{\min})}} \|c_1\| \|w^*(c_1) - w^*(c_2)\|^2.$$

*Moreover, if $c_1, c_2 \neq 0$, we have*

$$\|w^*(c_1) - w^*(c_2)\| \leq \frac{\sqrt{2L(r - g_{\min})}}{\mu} \cdot \left\| \frac{c_1}{\|c_1\|} - \frac{c_2}{\|c_2\|} \right\|.$$

Together with the following lemma, we can obtain a sharper bound on the inner product, compared to the one we previously obtained using the Cauchy-Schwarz inequality.

**Lemma C.8.** *Let $y_1, y_2 \in \mathbb{R}^d$ be non-zero vectors. Assume there exists a constant $C > 0$ such that $\|y_1\| \geq C$. Then, the following inequality holds:*

$$\left\| \frac{y_1}{\|y_1\|} - \frac{y_2}{\|y_2\|} \right\| \leq \frac{2}{C} \|y_1 - y_2\|$$

*Proof of Lemma C.8.* Consider the difference between the normalized vectors:

$$\Delta = \frac{y_1}{\|y_1\|} - \frac{y_2}{\|y_2\|}.$$

We add and subtract the term $\frac{y_2}{\|y_1\|}$ to facilitate factorization:

$$\Delta = \frac{y_1}{\|y_1\|} - \frac{y_2}{\|y_1\|} + \frac{y_2}{\|y_1\|} - \frac{y_2}{\|y_2\|}.$$

Grouping the terms, we obtain:

$$\Delta = \frac{1}{\|y_1\|}(y_1 - y_2) + y_2 \left( \frac{1}{\|y_1\|} - \frac{1}{\|y_2\|} \right).$$

Applying the triangle inequality yields:

$$\|\Delta\| \leq \frac{1}{\|y_1\|}\|y_1 - y_2\| + \|y_2\| \left| \frac{1}{\|y_1\|} - \frac{1}{\|y_2\|} \right|.$$

Next, we bound the scalar term. Finding a common denominator and applying the reverse triangle inequality $\|\|y_2\| - \|y_1\|\| \leq \|y_1 - y_2\|$:

$$\left| \frac{1}{\|y_1\|} - \frac{1}{\|y_2\|} \right| = \frac{\|\|y_2\| - \|y_1\|\|}{\|y_1\|\|y_2\|} \leq \frac{\|y_1 - y_2\|}{\|y_1\|\|y_2\|}.$$

Substituting this back into our expression for $\|\Delta\|$:

$$\|\Delta\| \leq \frac{\|y_1 - y_2\|}{\|y_1\|} + \|y_2\| \frac{\|y_1 - y_2\|}{\|y_1\|\|y_2\|}.$$

Canceling $\|y_2\|$ in the second term:

$$\|\Delta\| \leq \frac{\|y_1 - y_2\|}{\|y_1\|} + \frac{\|y_1 - y_2\|}{\|y_1\|} = \frac{2}{\|y_1\|}\|y_1 - y_2\|.$$

Given the lower bound assumption $\|y_1\| \geq C$, it follows that $\frac{1}{\|y_1\|} \leq \frac{1}{C}$. Therefore:

$$\left\| \frac{y_1}{\|y_1\|} - \frac{y_2}{\|y_2\|} \right\| \leq \frac{2}{C}\|y_1 - y_2\|.$$

$\square$

Equipped with these two lemmas, we are ready to prove a calibration bound.

*Proof of Corollary 3.9.* Let $f^*(x) = \frac{\mathbb{E}_P[Y|X=x]}{\mathbb{E}_P[\|Y\||X=x]}$ denote the Bayes risk minimizer of $\ell_{\text{WISE}}$. Follow the same analysis in Proposition 3.5, we get

$$\begin{aligned} R_P^{\ell_{\text{decision}}}(f) - R_P^* &= \mathbb{E}_P \left[ Y^\top w^*(f(X)) - Y^\top w^*(\mathbb{E}_P[Y|X]) \right] \\ &= \mathbb{E}_P \left[ \mathbb{E}_P[Y \mid X]^\top (w^*(f(X)) - w^*(\mathbb{E}_P[Y|X])) \right] \\ &= \mathbb{E}_P \left[ \mathbb{E}_P[\|Y\| \mid X] \frac{\mathbb{E}_P[Y \mid X]^\top}{\mathbb{E}_P[\|Y\| \mid X]} (w^*(f(X)) - w^*(\mathbb{E}_P[Y|X])) \right] \\ &= \mathbb{E}_P \left[ \mathbb{E}_P[\|Y\| \mid X] f^*(X)^\top (w^*(f(X)) - w^*(f^*(X))) \right]. \end{aligned}$$

We now invoke the properties of the strongly convex level sets established in Lemma C.7. Specifically, we apply the first inequality of Lemma C.7 with $c_1 = f^*(X)$ and $c_2 = f(X)$. This yields:

$$f^*(X)^\top (w^*(f(X)) - w^*(f^*(X))) \leq \frac{L}{2\sqrt{2\mu(r - g_{\min})}}\|f^*(X)\|\|w^*(f^*(X)) - w^*(f(X))\|^2.$$

Substituting this upper bound back into the expectation:

$$R_P^{\ell_{\text{decision}}}(f) - R_P^* \leq \mathbb{E}_P \left[ \mathbb{E}_P[\|Y\| \mid X] \frac{L}{2\sqrt{2\mu(r - g_{\min})}}\|f^*(X)\|\|w^*(f^*(X)) - w^*(f(X))\|^2 \right].$$

Next, we apply the second inequality of Lemma C.7 to bound the distance between the optimal decisions in terms of the angular difference of the cost vectors:

$$\|w^*(f^*(X)) - w^*(f(X))\| \leq \frac{\sqrt{2L(r - g_{\min})}}{\mu} \left\| \frac{f^*(X)}{\|f^*(X)\|} - \frac{f(X)}{\|f(X)\|} \right\|.$$

Squaring this term and combining it with the previous inequality, we obtain:

$$R_P^{\ell_{\text{decision}}}(f) - R_P^* \leq \mathbb{E}_P \left[ \mathbb{E}_P[\|Y\| \mid X] \|f^*(X)\| \frac{L^2 \sqrt{r - g_{\min}}}{\mu^{2.5}\sqrt{2}} \left\| \frac{f^*(X)}{\|f^*(X)\|} - \frac{f(X)}{\|f(X)\|} \right\|^2 \right].$$

Under Assumption 3.1 and the assumption that $\|\mathbb{E}_P[Y|X]\| \geq \gamma > 0$, we have $f^*(X)$ and $f(X)$ are both non-zero almost surely. Moreover, we have $\|f^*(X)\| = \frac{\|\mathbb{E}[Y|X]\|}{\mathbb{E}[\|Y\||X]} \geq \frac{\gamma}{B_y}$. We apply Lemma C.8 to relate the angular difference to the Euclidean distance:

$$\left\| \frac{f^*(X)}{\|f^*(X)\|} - \frac{f(X)}{\|f(X)\|} \right\|^2 \leq \frac{4B_y^2}{\gamma^2} \|f^*(X) - f(X)\|^2.$$

Note that $\|f^*(X)\| \leq 1$ by Jensen's inequality. Combining these results, we can bound the decision regret by the weighted squared error:

$$R_P^{\ell_{\text{decision}}}(f) - R_P^* \leq \frac{4B_y^2}{\gamma^2} \frac{L^2 \sqrt{r - g_{\min}}}{\mu^{2.5}\sqrt{2}} \mathbb{E}_P \left[ \mathbb{E}_P[\|Y\| \mid X] \|f^*(X) - f(X)\|^2 \right].$$

Finally, using the equivalent form of the excess WISE risk established in Lemma C.1, we conclude:

$$R_P^{\ell_{\text{decision}}}(f) - R_P^* \leq M(R_P^{\ell_{\text{WISE}}}(f) - R_P^{\ell_{\text{WISE}}}(f^*)),$$

where $M = \frac{2B_y^2 L^2 \sqrt{2(r - g_{\min})}}{\gamma^2 \mu^{2.5}}$.

Regarding the excess regret bound, we recall the bound for $\ell_{\text{WISE}}$ risk in equation (11).

To obtain the stated confidence form, set $\varepsilon = C_1'\delta^\nu$ in the preceding bound. Then the bound holds with probability at least $1 - C_1'\delta^\nu$ and

$$\nu \log(C_1 n) + \log(1/\varepsilon) = \nu \log(C_1 n) + \nu \log(1/\delta) - \log C_1'.$$

Since $\delta \leq (nd + 1)^{-C_0}$, choosing $C_0$ sufficiently large allows the term $\nu \log(C_1 n)$ to be absorbed into a constant multiple of $\nu \log(1/\delta)$. Renaming constants yields

$$R_P^{\ell_{\text{decision}}}(\hat{f}_n) - R_P^\star \leq C_2 \frac{\nu \log(1/\delta)}{n}$$

with probability at least $1 - C_1\delta^\nu$. $\qquad\qquad\square$

# D. Experimental Details

For our numerical experiment implementation, we adopted the PyEPO framework (Tang & Khalil, 2024b) where Gurobi is used as the optimization solver. For each method, we trained for 100 epochs and repeated for 20 independent trials to obtain the experiment result. The ADAM optimizer is used for learning predictors from a linear hypothesis class. Regarding hyperparameter configuration, we set $\lambda = 100$ for DBB (Pogančić et al., 2019), and $M = 3, \epsilon = 1.0$ for PFY (Berthet et al., 2020).

To evaluate the excess regret, we generate 2000 samples for the 2D knapsack problem, 10000 samples for the shortest path experiment, and 5000 samples for the portfolio problem.

Experiments were run on a Windows computer with an Intel(R) Core(TM) i9-14900F CPU.

### D.1. 2D Knapsack

In this experiment, the learning rate is set to be 1e-2 for $n = 100$, 5e-3 for $n = 200$, and 3e-3 for $n = 400, 800$. For a given cost vector $y$, the optimization formulation is:

$$\max_{\mathbf{w} \in \mathbb{R}^d} \quad \sum_{j=1}^{d} w_j y_j$$

$$\text{s.t.} \quad \sum_{j=1}^{d} w_j W_{1j} \leq W,$$

$$\sum_{j=1}^{d} w_j W_{2j} \leq W,$$

$$w_j \in \{0,1\}, j = 1, \ldots, d.$$

Table 1 summarizes the results of the experiment.

*Table 1.* 2D knapsack: normalized testing set regret (%) under different misspecification degrees and sample sizes. Mean $\pm$ Standard Deviation.

| Degree | Samples | MSE | WISE | SPO+ | PFY | DBB | CAVE | LAVA |
|--------|---------|-----|------|------|-----|-----|------|------|
| 1 | 100 | $12.43 \pm 0.39$ | $\mathbf{12.01 \pm 0.34}$ | $12.59 \pm 0.44$ | $13.45 \pm 0.53$ | $17.30 \pm 1.19$ | $23.02 \pm 4.12$ | $14.84 \pm 0.82$ |
| | 200 | $12.11 \pm 0.35$ | $\mathbf{11.35 \pm 0.19}$ | $11.93 \pm 0.33$ | $12.51 \pm 0.36$ | $16.28 \pm 1.93$ | $22.62 \pm 2.84$ | $13.27 \pm 0.49$ |
| | 400 | $11.71 \pm 0.22$ | $\mathbf{11.05 \pm 0.13}$ | $11.64 \pm 0.16$ | $12.32 \pm 0.19$ | $15.25 \pm 0.99$ | $19.54 \pm 2.58$ | $12.55 \pm 0.41$ |
| | 800 | $11.15 \pm 0.13$ | $\mathbf{11.05 \pm 0.11}$ | $11.53 \pm 0.17$ | $11.84 \pm 0.15$ | $14.24 \pm 1.06$ | $20.09 \pm 2.01$ | $11.49 \pm 0.21$ |
| 2 | 100 | $8.74 \pm 0.26$ | $\mathbf{8.32 \pm 0.22}$ | $9.09 \pm 0.35$ | $10.32 \pm 0.52$ | $17.11 \pm 2.90$ | $17.97 \pm 2.91$ | $11.88 \pm 0.64$ |
| | 200 | $8.65 \pm 0.30$ | $\mathbf{7.80 \pm 0.11}$ | $8.41 \pm 0.19$ | $9.21 \pm 0.30$ | $14.83 \pm 2.56$ | $16.78 \pm 2.10$ | $10.34 \pm 0.41$ |
| | 400 | $8.23 \pm 0.18$ | $\mathbf{7.63 \pm 0.10}$ | $8.05 \pm 0.13$ | $8.60 \pm 0.17$ | $12.28 \pm 1.67$ | $16.42 \pm 2.94$ | $9.08 \pm 0.30$ |
| | 800 | $\mathbf{7.80 \pm 0.09}$ | $7.85 \pm 0.09$ | $7.93 \pm 0.13$ | $8.49 \pm 0.17$ | $10.60 \pm 1.01$ | $15.14 \pm 1.62$ | $8.37 \pm 0.25$ |
| 4 | 100 | $\mathbf{6.64 \pm 0.42}$ | $6.89 \pm 0.37$ | $6.66 \pm 0.31$ | $7.58 \pm 0.35$ | $19.20 \pm 3.94$ | $11.17 \pm 1.66$ | $9.57 \pm 0.76$ |
| | 200 | $6.29 \pm 0.29$ | $6.43 \pm 0.22$ | $\mathbf{6.01 \pm 0.12}$ | $6.66 \pm 0.34$ | $14.69 \pm 3.87$ | $10.92 \pm 2.03$ | $8.10 \pm 0.56$ |
| | 400 | $5.88 \pm 0.20$ | $5.99 \pm 0.21$ | $\mathbf{5.55 \pm 0.17}$ | $5.90 \pm 0.14$ | $11.50 \pm 2.55$ | $10.30 \pm 2.31$ | $6.78 \pm 0.33$ |
| | 800 | $6.50 \pm 0.18$ | $6.42 \pm 0.14$ | $\mathbf{5.72 \pm 0.12}$ | $5.92 \pm 0.14$ | $9.59 \pm 2.61$ | $11.80 \pm 1.44$ | $6.09 \pm 0.22$ |

### D.2. Shortest Path

In this experiment, the learning rate is set to be 5e-3 for $n = 200, 400$, 2e-3 for $n = 800$, and 1e-3 for $n = 1600$. Regarding the data generating process, the input features are sampled as $x_i \overset{\text{i.i.d.}}{\sim} \mathcal{N}(0, I_5)$. The edge cost follows $y_{ij} = f_{ij}^*(x_i)(1 + \epsilon_{ij})$ where $\epsilon_{ij} \overset{\text{i.i.d.}}{\sim} \text{Unif}[-0.5, 0.5]$ and $f_{ij}$ follows

$$f_{ij}^*(x) = \frac{1}{3.5^{degree}} \left[ \left( \frac{1}{\sqrt{5}}(B^* x_i)_j + 3 \right)^{degree} + 1 \right]$$

where $B^* \in \{0,1\}^{40 \times 5}$ has i.i.d. Bernoulli$(0.5)$ entries.

For a given cost $y$, the optimization problem is formulated as follows:

$$\min_{w} \sum_{(i,j)\in E} y_{ij} w_{ij}$$

$$\text{Subject to:} \quad \sum_{j:(s,j)\in E} w_{sj} - \sum_{j:(j,s)\in E} w_{js} = 1 \qquad \text{(Source Node)}$$

$$\sum_{j:(i,j)\in E} w_{ij} - \sum_{j:(j,i)\in E} w_{ji} = 0 \qquad \forall i \in V \setminus \{s,t\}$$

$$\sum_{j:(t,j)\in E} w_{tj} - \sum_{j:(j,t)\in E} w_{jt} = -1 \qquad \text{(Sink Node)}$$

$$w_{ij} \in \{0,1\} \qquad \forall (i,j) \in E,$$

where $G = (V,E)$ is a directed graph representing the $5 \times 5$ grid, consisting of a set of nodes $V$ and edges $E$. We define $s$ as the source node and $t$ as the destination (sink) node. Note that the integer constraint can be relaxed since the constraint matrix is unimodular, and thus this problem can be solved in polynomial time (Wolsey & Nemhauser, 1999).

Figure 7 with Table 2 together illustrates a comprehensive result compared to Figure 4.

*Table 2.* Shortest path: normalized testing set regret (%) and total training time (s) under varying misspecification degrees and number of samples. Mean $\pm$ Standard Deviation.

| Degree | Samples | MSE | | WISE | | SPO+ | | PFY | | DBB | | LAVA | |
|---|---|---|---|---|---|---|---|---|---|---|---|---|---|
| | | Regret(%) | Time(s) | Regret(%) | Time(s) | Regret(%) | Time(s) | Regret(%) | Time(s) | Regret(%) | Time(s) | Regret(%) | Time(s) |
| 1 | 200 | **16.10 ± 0.57** | **1.42 ± 0.08** | 16.20 ± 0.60 | 1.58 ± 0.16 | 16.78 ± 0.63 | 130.02 ± 5.73 | 17.52 ± 0.69 | 172.65 ± 8.83 | 23.18 ± 1.75 | 267.73 ± 17.20 | 19.44 ± 0.95 | 6.45 ± 0.63 |
| | 400 | **15.81 ± 0.58** | **2.94 ± 0.36** | 15.93 ± 0.61 | 2.98 ± 0.22 | 16.18 ± 0.59 | 265.87 ± 8.25 | 16.73 ± 0.55 | 347.94 ± 20.23 | 21.22 ± 1.46 | 530.57 ± 14.70 | 17.82 ± 0.70 | 12.30 ± 1.63 |
| | 800 | **15.65 ± 0.58** | 6.04 ± 0.73 | 15.70 ± 0.58 | **5.78 ± 0.25** | 15.83 ± 0.61 | 520.57 ± 16.80 | 16.18 ± 0.60 | 713.45 ± 37.35 | 20.28 ± 1.73 | 1063.61 ± 30.35 | 17.08 ± 0.73 | 25.43 ± 2.33 |
| | 1600 | **15.53 ± 0.59** | **11.70 ± 2.03** | 15.62 ± 0.57 | 11.86 ± 1.19 | 15.63 ± 0.58 | 986.79 ± 132.03 | 15.82 ± 0.60 | 1378.56 ± 14.25 | 19.74 ± 1.53 | 2069.74 ± 40.72 | 16.72 ± 0.73 | 49.06 ± 5.27 |
| 2 | 200 | **10.96 ± 0.77** | **1.46 ± 0.10** | 11.01 ± 0.81 | 1.53 ± 0.10 | 11.52 ± 0.69 | 130.92 ± 3.23 | 12.04 ± 0.72 | 174.03 ± 6.23 | 25.04 ± 3.54 | 257.43 ± 13.50 | 14.09 ± 1.07 | 6.41 ± 0.48 |
| | 400 | **10.81 ± 0.79** | 2.97 ± 0.25 | 10.83 ± 0.81 | **2.93 ± 0.28** | 11.07 ± 0.80 | 271.62 ± 15.09 | 11.38 ± 0.79 | 354.23 ± 17.21 | 22.18 ± 4.47 | 526.84 ± 13.11 | 12.62 ± 0.94 | 12.57 ± 2.85 |
| | 800 | 10.73 ± 0.73 | 5.84 ± 0.80 | **10.69 ± 0.74** | **5.78 ± 0.51** | 10.82 ± 0.76 | 526.81 ± 19.07 | 10.94 ± 0.75 | 709.77 ± 25.45 | 18.51 ± 3.60 | 1042.49 ± 16.14 | 11.51 ± 0.86 | 21.40 ± 2.69 |
| | 1600 | **10.61 ± 0.75** | **11.45 ± 0.99** | 10.63 ± 0.74 | 11.47 ± 0.37 | 10.63 ± 0.75 | 1020.61 ± 56.45 | 10.74 ± 0.73 | 1366.48 ± 52.90 | 16.40 ± 1.64 | 2110.98 ± 120.82 | 11.30 ± 0.81 | 49.06 ± 5.95 |
| 4 | 200 | 9.29 ± 0.99 | **1.46 ± 0.15** | **8.48 ± 0.98** | 1.52 ± 0.04 | 8.82 ± 0.94 | 128.60 ± 7.44 | 8.92 ± 1.00 | 159.06 ± 5.88 | 39.21 ± 12.40 | 260.67 ± 13.43 | 13.52 ± 1.74 | 5.79 ± 1.08 |
| | 400 | 9.09 ± 1.05 | **2.97 ± 0.37** | 8.26 ± 1.06 | 2.98 ± 0.20 | **8.11 ± 0.91** | 251.47 ± 9.83 | 8.23 ± 0.84 | 350.64 ± 14.55 | 25.70 ± 5.21 | 523.71 ± 29.44 | 9.37 ± 0.91 | 12.73 ± 1.59 |
| | 800 | 8.99 ± 0.95 | **5.60 ± 0.46** | 8.09 ± 0.89 | 5.70 ± 0.45 | 7.81 ± 0.91 | 512.28 ± 25.68 | **7.79 ± 0.88** | 707.22 ± 38.22 | 24.30 ± 9.89 | 1003.85 ± 41.60 | 8.41 ± 0.95 | 27.01 ± 4.04 |
| | 1600 | 8.79 ± 0.99 | **10.79 ± 0.43** | 8.03 ± 0.90 | 11.79 ± 1.15 | 7.64 ± 0.84 | 1007.58 ± 76.91 | **7.62 ± 0.83** | 1363.92 ± 47.81 | 17.08 ± 3.38 | 2007.60 ± 143.52 | 8.11 ± 0.93 | 47.54 ± 8.62 |
| 6 | 200 | 14.15 ± 2.09 | **1.53 ± 0.18** | 10.76 ± 1.56 | 1.62 ± 0.15 | **9.74 ± 1.29** | 133.37 ± 7.28 | 9.79 ± 1.32 | 171.14 ± 13.53 | 55.02 ± 19.27 | 259.30 ± 23.50 | 17.95 ± 3.16 | 6.54 ± 0.73 |
| | 400 | 13.49 ± 2.02 | **2.88 ± 0.26** | 10.09 ± 1.59 | 2.90 ± 0.10 | **8.76 ± 1.36** | 262.18 ± 16.30 | 8.77 ± 1.36 | 357.57 ± 15.06 | 32.49 ± 9.77 | 507.86 ± 35.04 | 10.89 ± 1.49 | 12.05 ± 3.02 |
| | 800 | 13.25 ± 1.84 | **5.42 ± 0.28** | 9.74 ± 1.39 | 5.66 ± 0.41 | **8.11 ± 1.19** | 501.70 ± 32.86 | 8.11 ± 1.19 | 684.61 ± 25.87 | 28.35 ± 9.50 | 971.81 ± 57.21 | 8.94 ± 1.29 | 24.96 ± 1.48 |
| | 1600 | 12.78 ± 2.05 | 12.16 ± 2.23 | 9.76 ± 1.39 | **11.66 ± 0.92** | **7.99 ± 1.15** | 1072.66 ± 68.49 | 7.99 ± 1.18 | 1393.62 ± 65.80 | 21.81 ± 7.43 | 1972.05 ± 198.86 | 8.63 ± 1.32 | 50.83 ± 3.97 |
| 8 | 200 | 23.94 ± 3.73 | **1.46 ± 0.14** | 16.87 ± 3.26 | 1.53 ± 0.12 | 12.69 ± 2.21 | 126.56 ± 4.21 | **12.67 ± 2.35** | 172.21 ± 10.58 | 78.63 ± 27.91 | 245.07 ± 12.48 | 25.86 ± 5.35 | 5.73 ± 0.78 |
| | 400 | 23.26 ± 3.84 | **2.77 ± 0.22** | 15.17 ± 2.93 | 2.93 ± 0.10 | **11.30 ± 2.16** | 272.46 ± 15.96 | 11.33 ± 2.25 | 354.74 ± 24.27 | 41.58 ± 13.10 | 505.25 ± 19.67 | 14.84 ± 2.37 | 13.14 ± 1.39 |
| | 800 | 22.44 ± 4.12 | **5.44 ± 0.43** | 14.45 ± 2.59 | 5.67 ± 0.18 | 10.63 ± 2.03 | 541.43 ± 35.77 | **10.55 ± 2.10** | 688.41 ± 22.76 | 40.89 ± 23.80 | 990.40 ± 57.33 | 12.37 ± 3.11 | 22.30 ± 3.27 |
| | 1600 | 21.62 ± 4.27 | **11.16 ± 0.57** | 14.55 ± 2.50 | 12.57 ± 3.07 | **10.33 ± 1.95** | 1047.50 ± 33.77 | 10.37 ± 1.96 | 1379.46 ± 69.33 | 29.24 ± 9.40 | 1994.49 ± 136.90 | 11.49 ± 2.37 | 48.53 ± 3.85 |

### D.3. Portfolio Optimization

We considered feature dimension $p = 6$, number of stocks $d = 25$. Regarding the data generating process, the input features are sampled as $x_i \overset{\text{i.i.d.}}{\sim} \mathcal{N}(0, I_p)$. The asset returns follow an additive structure $r_i = \mu(x_i) + \mathbf{e}_i$, where the conditional mean $\mu(x_i)$ is defined component-wise as:

$$\mu_j(x_i) = \left( \frac{0.05}{\sqrt{p}} (Bx_i)_j + 0.2^{1/degree} \right)^{degree}$$

where $B \in \{0,1\}^{d \times p}$ contains i.i.d. Bernoulli$(0.5)$ entries. The random noise term $\mathbf{e}_i$ captures heavy-tailed behavior and correlations through a factor model:

$$\mathbf{e}_i = L\boldsymbol{\zeta}_i + 0.01\boldsymbol{\epsilon}_i$$

where $L \in \mathbb{R}^{d \times p}$ is a random loading matrix with each entry sampled from $\text{Unif}[-0.0025, 0.0025]$, and the components $\boldsymbol{\zeta}_i \in \mathbb{R}^p$ and $\boldsymbol{\epsilon}_i \in \mathbb{R}^d$ are drawn independently from a standardized Student's $t$-distribution with degrees of freedom $\nu = 3$ (scaled to satisfy $\text{Var}(\cdot) = 1$). Let $\Sigma = LL^\top + (0.01)^2 I_{25}$ denote the covariance matrix of the stocks.

Then the optimization formulation is:

*Figure 7.* Average normalized test regret vs average training time for the shortest path problem, under varying degrees of misspecification and numbers of training samples. Results are collected from 20 trials and 100 training epochs.

$$\max_{\mathbf{w} \in \mathbb{R}^d} \quad \sum_{i=1}^{d} w_i r_i$$
$$\text{s.t.} \quad \mathbf{w}^\top \mathbf{1} \leq 1,$$
$$\mathbf{w}^\top \Sigma \mathbf{w} \leq \gamma,$$
$$\mathbf{w} \geq 0,$$

where $\gamma = 2.25 \cdot \frac{\sum_{i,j} \Sigma_{i,j}}{25^2}$ is a given risk level tolerance.

Table 3 summarizes the results, where we can see the gap between different methods is narrower compared to the previous two experiments.

*Table 3.* Portfolio optimization: normalized testing set regret (%) under different misspecification degrees and sample sizes. Mean ± Standard Deviation.

| Degree | Samples | MSE | WISE | SPO+ | PFY | DBB |
|---|---|---|---|---|---|---|
| 1 | 200 | **0.88 ± 0.02** | 0.89 ± 0.02 | 0.96 ± 0.03 | 0.88 ± 0.02 | 0.95 ± 0.04 |
| | 400 | **0.86 ± 0.02** | 0.87 ± 0.02 | 0.91 ± 0.03 | 0.87 ± 0.02 | 0.89 ± 0.02 |
| | 800 | **0.85 ± 0.02** | 0.86 ± 0.02 | 0.89 ± 0.02 | 0.86 ± 0.02 | 0.87 ± 0.02 |
| | 1600 | **0.85 ± 0.02** | 0.86 ± 0.02 | 0.87 ± 0.02 | 0.85 ± 0.02 | 0.86 ± 0.02 |
| 3 | 200 | 0.84 ± 0.02 | **0.83 ± 0.02** | 0.91 ± 0.03 | 0.83 ± 0.02 | 0.91 ± 0.04 |
| | 400 | 0.82 ± 0.02 | **0.81 ± 0.02** | 0.86 ± 0.03 | 0.82 ± 0.02 | 0.83 ± 0.02 |
| | 800 | 0.81 ± 0.02 | **0.80 ± 0.02** | 0.84 ± 0.02 | 0.81 ± 0.02 | 0.82 ± 0.02 |
| | 1600 | 0.81 ± 0.02 | 0.80 ± 0.02 | 0.82 ± 0.02 | **0.80 ± 0.02** | 0.81 ± 0.02 |
| 7 | 200 | 0.56 ± 0.03 | **0.48 ± 0.02** | 0.51 ± 0.02 | 0.48 ± 0.02 | 0.52 ± 0.02 |
| | 400 | 0.53 ± 0.02 | **0.46 ± 0.02** | 0.48 ± 0.02 | 0.48 ± 0.02 | 0.47 ± 0.02 |
| | 800 | 0.52 ± 0.02 | **0.46 ± 0.01** | 0.47 ± 0.01 | 0.47 ± 0.01 | 0.46 ± 0.01 |
| | 1600 | 0.52 ± 0.02 | **0.45 ± 0.02** | 0.46 ± 0.02 | 0.46 ± 0.02 | 0.45 ± 0.01 |

