# OpenReview forum: "A Solver-Free Training Method for Predict-then-Optimize"
_ICML.cc/2026/Conference — ICML 2026 regular_

### Official Review · Reviewer_rf1r · 2026-02-15

**Soundness:** 3
**Presentation:** 2
**Significance:** 3
**Originality:** 2
**Overall Recommendation:** 4
**Confidence:** 3

**Summary:**

This paper addresses the computational bottleneck in the so-called "predict-then-optimize" setting. In contrast to costly existing decision-focused methods, which require expensive solver calls, the authors propose a solver-free training objective named WISE (Weight Integrated Spherical Error). This method exploits the scale invariance and loss scaling properties of linear optimization, thereby incorporating the geometry of optimization problems into the training process, while avoiding the need for solver calls.

Theoretically, the authors prove that WISE is Fisher consistent with the decision regret, and also provide excess risk bounds following the setup of prior work. Finally, experiments demonstrate that WISE achieves competitive or superior decision quality compared to existing methods while being as fast as the standard two-stage method.

**Compliance With Llm Reviewing Policy:**

Affirmed.

**Final Justification:**

The rebuttal addressed my concern regarding the comparison with LAVA (Berden et al., NeurIPS 2025), which led me to update my score from 3 to 4. The reason why I maintain a weak accept rather than a stronger recommendation is that the paper lacks a rigorous theoretical advantage over existing fast-rate results, e.g., Hu et al.

**Key Questions For Authors:**

Q1. How does WISE compare with the method proposed by Berden et al. (NeurIPS 2025) in the linear program setting (e.g., the shortest path problem)?

Q2. Is it possible to establish a fast rate of WISE under the low-noise condition, similar to Hu et al. Furthermore, is there any theoretical advantage of WISE over the standard two-stage method?

Q3. In the knapsack and shortest path experiments, WISE's performance degrades as the misspecification level increases, compared to SPO+ and PFY. Can you explain why this happens? In particular, is there any insight into how solver-free and solver-based methods are affected differently by misspecification?

Minor comment:
In Figure 3, the polynomial degrees shown in the images and the caption are inconsistent.

**Limitations:**

The proposed method requires labels of objective coefficients $y$ for training, which may not be available in many real-world applications. In contrast, papers in related areas, such as Mishra et al. (ICML 2024) "From Inverse Optimization to Feasibility to ERM" and references therein, only require observing solutions of optimization problems. This limitation should be clearly discussed.

**Strengths And Weaknesses:**

S1. The paper presents an efficient solver-free method that would be useful in practice. It is also easy to implement, given that datasets include complete information of objective coefficients $y$ and contextual features $x$.

S2. The paper is well-written, and the main concept is clearly explained.

W1. The paper is missing a comparison with a relevant work: Berden et al. (NeurIPS 2025) "Solver-Free Decision-Focused Learning for Linear Optimization Problems." While their method is specialized for linear programs, the "solver-free" concept is close to the current paper, and it is important to compare the two methods in terms of both decision quality and computational efficiency.

W2. Regarding the excess risk bound (Theorem 3.7), Hu et al. establish a fast rate of MSE under the low-noise condition (without strong convexity of level sets). This is not mentioned when discussing Theorem 3.7, and it is only stated that Theorem 3.7 matches the classic $O(n^{-1/2})$ order. Given this, the theoretical advantage of WISE over the standard two-stage method is not clear.

---

> ### Author Rebuttal · Authors · 2026-03-30
>
> **Re W1 & Q1: Missing Benchmarks**
>
> Following your valuable suggestion, LAVA is added as baselines to the 2D Knapsack and shortest path experiments. Due to the space limit, tables and figures are provided in [https://anonymous.4open.science/r/ICML2026_rebuttal-530B/README.md ]
>
> * In the 2D Knapsack experiment, WISE maintains an advantage in low-to-medium misspecification regimes (Degrees 1 and 2), yielding regret reductions of up to 3.5% over LAVA. Conversely, LAVA outperforms WISE in high misspecification settings (Degree 4 with 800 samples: 6.09% vs. 6.42%).
>
> * In the shortest path experiment, both methods show strong performance, revealing a distinct trade-off. Under low-to-medium misspecification (Degrees 1, 2, and 4), WISE improved regret by 1.5% to 5% compared to LAVA. Under high misspecification (Degrees 6 and 8), WISE maintains an advantage at smaller sample sizes (N=200), improving regret by 7.19% at Degree 6 and 8.99% at Degree 8. However, as the sample size scales, LAVA outperforms WISE (e.g., 11.49% vs. WISE's 14.55% at Degree 8, N=1600). While both methods avoid solver calls during gradient updates, WISE maintains a computational advantage. **Across all settings in the experiments, WISE is ~4x faster than LAVA in total training time.**
>
> These results confirm the high efficacy of both LAVA and WISE. Specifically, LAVA demonstrates robustness against high misspecification, whereas WISE offers a computational advantage when predictive power is high, thereby achieving a good balance between quality and efficiency.
>
>
> Besides the above numerical comparison, we will also include the following analytical comparison in the camera-ready version:
>
> > The LAVA method offers a highly elegant, solver-free approach for LPs, achieving both good computational efficiency and decision quality. Yet, its preprocessing step depends on having the ground-truth optimal solutions as input data. If the available dataset only provides the true cost vectors, a solver call is still needed during the preprocessing step before training can begin.
>
> > In contrast, our method works for general linear objective optimization problems regardless of the feasible region. Most importantly, **our approach is entirely solver-free during both the preprocessing phase and the training phase**. Our method also enjoys the Fisher consistency without restrictive assumptions, while the theoretical consistency is still under exploration for LAVA.
>
> **Re W2 & Q2: Convergence rates and theoretical advantages**
>
> We appreciate your insightful question. Our Theorem 3.7 was designed to match the classic $O(n^{-1/2})$ rates under standard assumptions. Indeed, there is a distinct theoretical advantage of WISE over the standard two-stage method under specific noise structures. For example, if noise is only over the magnitude and not the direction, WISE achieves a faster rate, since our loss absorbs magnitude variance via projection onto the unit sphere. Rigorous discussion of this advantage is left to ongoing works, but we value you pointing out this interesting direction.
>
> **Re Q3: degradation under high mis-specification**
>
> Our method can be viewed as a geometry-aware specialization of the two-stage method. As the hypothesis class becomes highly misspecified (e.g., trying to fit a linear model to highly non-linear data), standard two-stage and WISE methods struggle because they lack information from the specific optimization landscape. Solver-based methods like SPO+ or PFY call the oracle during training, effectively querying the problem structure. This active feedback acts as a corrective mechanism, aggressively penalizing predictions that cross critical decision boundaries, especially when the model is misspecified. On the other hand, completely solver-free methods naturally trade off this information for computational efficiency (e.g., a ~100x speedup in 5*5 shortest path).
>
> Hence, solver-based methods offer robustness against extreme misspecification, while solver-free methods (like WISE) are preferred when the model is reasonably well-specified, or for massive problem scales where a solver-free approach is the only practical choice.
>
> **Re Minor comment: Typo in Fig 3**
>
> Thank you for catching this. We have corrected the caption.
>
> **Re Limitations: Data structure requirement**
>
> We appreciate you pointing out the observability of the true parameter for training. We agree that only observing the true optimal solution is a relaxed assumption. The goal of our paper is to propose a Fisher consistent loss on the same space of cost vectors, identical to the setup in SPO+, PG, PFY, etc. Thus, WISE is suitable for the standard predict-then-optimize paradigm, assuming the availability of true cost $y$ during training. If only optimal decisions are available, Mishra et al. (2024) and Berden et al. (2025) would be better choices. We will properly include this limitation and compare with these papers in the camera-ready introduction.

---

> > ### Author Rebuttal · Reviewer_rf1r · 2026-03-31
> >
> > Thank you for the detailed response. The experimental comparisons with LAVA appear to be good. I encourage the author to discuss the degradation in the mis-specified case sufficiently in the paper. The lack of rigorous theoretical advantage compared to existing fast rate results still appears to be a weakness. Overall, the response has improved my impression, and I will update my score.

---

> > > ### Author Response · Authors · 2026-04-07
> > >
> > > We appreciate your feedback throughout the review and thank you for increasing your score.

---

### Official Review · Reviewer_F581 · 2026-03-12

**Soundness:** 3
**Presentation:** 3
**Significance:** 2
**Originality:** 2
**Overall Recommendation:** 4
**Confidence:** 3

**Summary:**

The paper proposes an optimization-solver-free method for scalable predict-then-optimize training by measure transformation. They propose a surrogate regret function the Weight Integrated Spherical Error (WISE), and propose a two-step transformation of the original probability measure: a reweighting based on cost magnitude, and a projection onto the unit sphere.They further provide some theoretical guarantees for the WISE loss, including Fisher consistency, calibration bounds, and finite sample excess regret bounds.

**Compliance With Llm Reviewing Policy:**

Affirmed.

**Final Justification:**

The explanation about handling unknowns in the constraints and the comparisons with LAVA are clear. My questions have been perfectly resolved, and I will raise my score.

**Key Questions For Authors:**

1.How easy is it to meet the assumptions that enable O(n^{−1}) convergence? Are these assumptions approximately satisfied in the experiments conducted?

**Limitations:**

See “Weaknesses”.

**Strengths And Weaknesses:**

Strengths:

1. The paper detects the scalability limitation of DFL, which is important to address.

2. The writing is generally clear.

3. The theoretical properties of the WISE loss are investigated and clearly stated.

Weaknesses:

1. The application scope of the proposed method is limited, only covering linear optimization problem with unknowns only in the objective.

2. All experiments are conducted on simulated datasets. However, realistic datasets for some problems, for example, knapsack problem, are available and widely used in DFL research.

3. The seleced baseline should include more recent works, for example, LAVA [1] is also a solver-free DFL approach and applicable for LP, experiment comparison with [1] would be beneficial.

[1] Berden, Senne, et al. "Solver-Free Decision-Focused Learning for Linear Optimization Problems." NeurIPS 2025.

---

> ### Author Rebuttal · Authors · 2026-03-30
>
> Due to the space limit, tables and figures of added experiments are provided in [https://anonymous.4open.science/r/ICML2026_rebuttal-530B/README.md ]
>
> **Re Weakness1: limited scope**
>
> Thank you for highlighting the scope of our method. While extending DFL to handle unknowns in the constraints is an important direction, it fundamentally changes the mathematical nature of the problem. If an unknown appears on the right-hand side of a constraint, achieving Fisher consistency using a point estimation becomes generally impossible. A recent paper [1] discussed the sufficient and necessary conditions for a Fisher consistent point prediction under some special settings, demonstrating that a general unknown constraint setting is highly restrictive.
>
>
> **Re Weakness2: experiment on simulated dataset**
>
> While we acknowledge the value of real datasets, we prioritized simulated datasets to maintain consistency with classical literature and to enable an \textbf{extensive sensitivity analysis}. Real-world datasets provide a static snapshot of performance, whereas our synthetic data generation process allows us to explicitly control and vary the degree of model misspecification (e.g., by adjusting the polynomial degree of the true data-generating process ). This allows us to rigorously evaluate the robustness of our method against baselines under various conditions.
>
> **Re Weakness3: missing benchmarks**
>
> Following your valuable suggestion, LAVA is added as baselines to the 2D Knapsack and shortest path experiments.
>
> * In the 2D Knapsack experiment, WISE maintains an advantage in low-to-medium misspecification regimes (Degrees 1 and 2), yielding regret reductions of up to 3.5% over LAVA. Conversely, LAVA outperforms WISE in high misspecification settings (Degree 4 with 800 samples: 6.09% vs. 6.42%).
>
> * In the shortest path experiment, both methods show strong performance, revealing a distinct trade-off. Under low-to-medium misspecification (Degrees 1, 2, and 4), WISE improved regret by 1.5% to 5% compared to LAVA. Under high misspecification (Degrees 6 and 8), WISE maintains an advantage at smaller sample sizes (N=200), improving regret by 7.19% at Degree 6 and 8.99% at Degree 8. However, as the sample size scales, LAVA outperforms WISE (e.g., 11.49% vs. WISE's 14.55% at Degree 8, N=1600). While both methods avoid solver calls during gradient updates, WISE maintains a computational advantage. **Across all settings in the experiments, WISE is ~4x faster than LAVA in total training time.**
>
> These results confirm the high efficacy of both LAVA and WISE. Specifically, LAVA demonstrates robustness against high misspecification, whereas WISE offers a computational advantage when predictive power is high, thereby achieving a good balance between quality and efficiency.
>
> Besides the above numerical comparison, we will also include the following analytical comparison in the camera-ready version:
>
> > The LAVA method offers a highly elegant, solver-free approach for LPs, achieving both good computational efficiency and decision quality. Yet, its preprocessing step depends on having the ground-truth optimal solutions as input data. If the available dataset only provides the true cost vectors, a solver call is still needed during the preprocessing step before training can begin.
>
> > In contrast, our method works for general linear objective optimization problems regardless of the feasible region. Most importantly, **our approach is entirely solver-free during both the preprocessing phase and the training phase**. Our method also enjoys the Fisher consistency without restrictive assumptions, while the theoretical consistency is still under exploration for LAVA.
>
>
> **Re Question: assumptions for fast convergence rates**
>
> This assumption on strongly convex level set is first studied in [2]. In practice, this assumption is relatively straightforward to meet for specific classes of problems, such as those involving bounded $L_2$ norm balls or ellipsoids. In our experiments, this condition is indeed approximately satisfied by the Portfolio Optimization problem, which features a linear objective subject to quadratic risk constraints. As discussed in Appendix C1 of [2], the portfolio optimization setup is a slightly generalized version of having strongly convex level sets, ensuring the smooth transition between prediction and decision that enables these faster rates.
>
>
>
> * [1]: Er, C., & Liu, M. (2025). Decision-focused bias correction for fluid approximation. arXiv preprint arXiv:2512.15726.
> * [2]: Liu, H., & Grigas, P. (2021). Risk bounds and calibration for a smart predict-then-optimize method. Advances in Neural Information Processing Systems, 34, 22083-22094.

---

> > ### Author Rebuttal · Reviewer_F581 · 2026-04-03
> >
> > Thank you for the thorough rebuttal. The explanation about handling unknowns in the constraints and the comparisons with LAVA are clear. My questions have been perfectly resolved, and I will raise my score.

---

> > > ### Author Response · Authors · 2026-04-07
> > >
> > > Thank you for confirming that our response resolved your questions, and we appreciate your willingness to increase your evaluation.

---

### Official Review · Reviewer_2x3J · 2026-03-12

**Soundness:** 4
**Presentation:** 4
**Significance:** 4
**Originality:** 4
**Overall Recommendation:** 6
**Confidence:** 4

**Summary:**

The paper derives a new surrogate loss function in contextual stochastic optimization that is Fisher consistent and does not need an optimization solver for training. The paper derives an excess regret bound for the empirical risk minimizer. Finally, the paper demonstrates the effectiveness and computational efficiency of the proposed approach numerically.

**Compliance With Llm Reviewing Policy:**

Affirmed.

**Final Justification:**

The rebuttal has reinforced my prior assessment and addressed all my questions. The paper provides a fresh perspective to contextual stochastic optimization and I support the acceptance strongly (6: Strongly Accept).

**Key Questions For Authors:**

Do the measure transformation and the corresponding theoretical results hold the same if Steps 1 and 2 are swapped?

**Limitations:**

yes

**Strengths And Weaknesses:**

Soundness: the theoretical results and their proofs make sense to me.

Presentation: the writing is clear and well-articulated.

Significance: the paper addresses an important problem in optimization (contextual stochastic optimization) and the proposed concept is innovative and a departure from the main-stream, numerically challenging approaches.

Originality: I appreciate that the paper proposes a new, solver-free surrogate function that can be applied efficiently in implementation.

---

> ### Author Rebuttal · Authors · 2026-03-30
>
> We sincerely thank you for your positive feedback and for recognizing the theoretical soundness, originality, and practical efficiency of our solver-free approach.
>
> **Re Question: Swapping steps of transformation**
>
> Thank you for this insightful question about swapping the steps of transformation.
> However, the measure transformation and the corresponding theoretical results would not hold if the steps were swapped. The specific sequence—Step 1 (Reweighting) followed by Step 2 (Projection)—is mathematically necessary to maintain Fisher consistency. If we first apply the projection, all the datapoints will have the same magnitude, and hence the reweighting will place all datapoints with the same weights.

---

> > ### Author Rebuttal · Reviewer_2x3J · 2026-03-31
> >
> > Thanks for addressing my question. I will keep my score the same.

---

### Official Review · Reviewer_FTGh · 2026-03-13

**Soundness:** 3
**Presentation:** 4
**Significance:** 3
**Originality:** 2
**Overall Recommendation:** 3
**Confidence:** 4

**Summary:**

The authors propose a solver-free approach for producing decision-focused learning models. To do so, they introduce a two-step methodology that i. first, scales the data and, ii. projects it onto the unit ball.  They provide a well-grounded theory to show that these two steps are necessary to generate a prediction model with desirable properties. Based on this idea, they define a surrogate loss function called WISE, which is, to some extent, a weighted squared error measure. The authors establish Fisher consistency for this surrogate loss function and demonstrate the effectiveness of their method.

**Compliance With Llm Reviewing Policy:**

Affirmed.

**Final Justification:**

I updated my score accordingly to the responses. Thank you very much for your responses

**Key Questions For Authors:**

The assumption of the existence of a consistent tie-breaking rule it is necessary, however, once the rule is established to optimize considering this rule could be quite difficult?

Actually, many papers considers the pessimistic assumption (amongs all the way of selecting one optimal solution, w^* chooses the one that worsen the most the regret). What rule do you choose in your implementation?

What kind of model are you training with your approach? A linear one?

**Limitations:**

Yes

**Strengths And Weaknesses:**

The authors propose a solver-free approach for decision-focused learning. Specifically, they introduce a two-step methodology based on scaling the data and projecting it onto the unit ball. Building on this idea, they define a surrogate loss function called WISE, which can be interpreted, to some extent, as a weighted squared-error measure. The paper establishes Fisher consistency for this surrogate loss and provides computational evidence supporting the effectiveness of the approach.

Overall, the paper is technically sound, although I still have some concerns regarding the correctness of some proofs (see my questions to the authors).

I find the central idea of modifying the data in a principled way to train decision-focused models to be original and interesting. However, the paper is written in a way that suggests this is the first solver-free predict-then-optimize method, which is not accurate. There are already several related contributions in the literature. I recommend that the authors position their work more carefully with respect to prior research, including the following surveys:

Mandi, J., Kotary, J., Berden, S., Mulamba, M., Bucarey, V., Guns, T., & Fioretto, F. (2024). Decision-focused learning: Foundations, state of the art, benchmark and future opportunities. Journal of Artificial Intelligence Research, 80, 1623–1701.

Sadana, U., Chenreddy, A., Delage, E., Forel, A., Frejinger, E., & Vidal, T. (2025). A survey of contextual optimization methods for decision-making under uncertainty. European Journal of Operational Research, 320(2), 271–289.

In particular, the idea of solver-free approaches has already been discussed extensively in works such as:

Tang, B., & Khalil, E. B. (2024, May). CAVE: A cone-aligned approach for fast predict-then-optimize with binary linear programs. In International Conference on the Integration of Constraint Programming, Artificial Intelligence, and Operations Research (pp. 193–210). Springer Nature Switzerland.

Berden, S., Mahmutoğulları, A. İ., Tsouros, D., & Guns, T. Solver-Free Decision-Focused Learning for Linear Optimization Problems. In The Thirty-Ninth Annual Conference on Neural Information Processing Systems.

Furthermore, the proposed WISE loss function appears closely related to the losses introduced in

I therefore encourage the authors to better clarify the novelty of their contribution, especially in relation to existing solver-free approaches and surrogate losses.

---

> ### Author Rebuttal · Authors · 2026-03-30
>
> Due to the space limit, tables and figures of the added experiments are provided in [https://anonymous.4open.science/r/ICML2026_rebuttal-530B/README.md ]
>
> **Re Weakness: Positioning contributions**
>
> We sincerely thank you for pointing us to these highly relevant papers and surveys. We have properly cited these survey papers in the revised version, and carefully compared our proposed method with CAVE and LAVA.
>
> The detailed numerical performance comparison is provided in our next response. Here, we first present an analytical comparison of our proposed loss with LAVA and CAVE.
>
> > CaVE introduces an innovative framework for LP/BLP that replaces computationally heavy solver calls with projections. However, it still requires solving a QP during each iteration. Moreover, an initial optimization step is needed during preprocessing to obtain the binding constraints associated with each true cost vector.
>
> > LAVA also offers a highly elegant, solver-free approach for LP, achieving both good computational efficiency and decision quality. Yet, its preprocessing step depends on having the ground-truth optimal solutions as input data. If the available dataset only provides the true cost vectors, a full solver call is still required during the preprocessing step before training can begin.
>
> > In contrast, our method works for general linear objective problems regardless of the feasible region (e.g., a quadratic programming with a linear objective). Most importantly, **our approach is entirely solver-free during both the preprocessing phase and the training phase**. Our method also enjoys the Fisher consistency without restrictive assumptions, while the theoretical consistency is still under exploration for CAVE and LAVA.
>
> We apologize if our writing inadvertently gave the impression that we were the very first solver-free DFL method; **our writing only claims that we are the first to approach DFL specifically via a measure transformation principle, not the first DFL solver-free method**. We will revise our Introduction and Related Work sections to include the suggested surveys and contrast our contributions with CaVE and LAVA.
>
> **Re Weakness: Missing Benchmarks**
>
> Following your valuable suggestion, both CaVE and LAVA are added as baselines to the 2D Knapsack, and LAVA is added to the Shortest Path experiments.
>
> * In the 2D Knapsack experiment, WISE consistently outperforms CaVE across all 12 evaluated settings, yielding an average regret improvement of 7.7%. Comparing with LAVA, WISE maintains a slight advantage in low-to-medium misspecification regimes (Degrees 1 and 2), yielding regret reductions of up to 3.5% over LAVA. Conversely, LAVA outperforms WISE in high misspecification settings (Degree 4 with 800 samples: 6.09% vs. 6.42%).
>
> * In the shortest path experiment, both methods again show strong performance, revealing a distinct trade-off. Under low-to-medium misspecification (Degrees 1, 2, and 4), WISE improved regret by 1.5% to 5% compared to LAVA (e.g., at Degree 4 with N=200, 8.48% vs. LAVA's 13.52%). Under high misspecification (Degrees 6 and 8), WISE maintains an advantage at smaller sample sizes, improving regret by 7.19% at Degree 6 (N=200: 10.76% vs 17.95%) and 8.99% at Degree 8 (N=200: 16.87% vs. 25.86%). However, as the sample size scales, LAVA outperforms WISE (e.g., achieving 11.49% vs. WISE's 14.55% at Degree 8 with N=1600). While both methods avoid solver calls during the core gradient updates, WISE maintains a computational advantage. **Across all settings in the experiments, WISE is ~4x faster than LAVA in total training time.**
>
> These results confirm the high efficacy of both LAVA and WISE. Specifically, LAVA demonstrates robustness against high misspecification, whereas WISE offers a computational advantage when predictive power is high, thereby achieving a good balance between quality and efficiency.
>
> **Re Question: Tie-Breaking Rules**
>
> Thank you for bringing attention to the tie-breaking rules (such as the pessimistic assumption used in methods like SPO+). This assumption was only for theoretical analysis; **it is not a requirement for implementation**. Because our WISE surrogate loss is just a transformed squared loss, the training process completely bypasses the solver. Hence, we do not need such a tie-breaking rule during training. The solver is only called when evaluating the final performance; any valid tie-breaking rule will yield the same objective value, meaning the specific tie chosen has no impact on the final regret.
>
> **Re Question: Prediction models**
>
> We indeed consider learning from a linear prediction class, which is clarified in the second paragraph of the numerical experiment. The linear prediction class is commonly used in past literature. We chose linear models because they allow us to perform controlled and extensive sensitivity analyses regarding the impact of model misspecification on method performance.

---

> > ### Author Rebuttal · Reviewer_FTGh · 2026-03-31
> >
> > Even though I do not agree that the tie breaking rule is only a matter of theoretical analysis, I think that authors tackle my response in a proper way.

---

> > > ### Author Response · Authors · 2026-04-02
> > >
> > > Thank you for your follow-up and for acknowledging that your concerns have been resolved.
> > >
> > >
> > > During the *training* process, the tie-breaking rule is not needed, as our training loss is a transformed squared loss and is solver-free:
> > >
> > > $$l_{WISE}(\hat{y}, y) = \|y\| \cdot \left\|\hat{y} - \frac{y}{\|y\|} \right\|^2  \text{if } \|y\| \neq 0,
> > >  l_{WISE}(\hat{y}, y) = 0  \text{ if } \|y\| = 0.
> > > $$
> > > When evaluating performance empirically, as you rightly noted, the presence of multiple optimal solutions suggests that a deliberately designed tie-breaking rule applied to the solver could further enhance robustness. Since the main focus of the paper is on the training method, and the evaluation procedure does not affect the theoretical analysis, any fixed tie-breaking rule can be adopted without loss of generality.
> > >
> > > Finally, we sincerely thank you for taking the time to engage with our rebuttal, and we are glad that our responses have resolved your concerns. We hope this positive resolution will be reflected in your final assessment of the paper.

---

### Decision · Program_Chairs · 2026-04-30

**Decision:**

Accept (regular)

**Comment:**

This paper introduces a scalable, optimization-solver-free surrogate loss  for decision-focused learning.
I recommend for acceptance since during the review and rebuttal phase it was agreed that the approach is original, theoretically grounded, and practical: it provides substantial speedups over standard pipelines while maintaining mathematically rigorous guarantees.